# Semi-Supervised Noise Adaptation: Transferring Knowledge from Noise Domain

## Abstract

Transfer learning aims to facilitate the learning of a target domain by transferring knowledge from a source domain. The source domain typically contains semantically meaningful samples (*e.g.*, images) to facilitate effective knowledge transfer. However, a recent study observes that the noise domain constructed from simple distributions (*e.g.*, Gaussian distributions) can serve as a surrogate source domain in the semi-supervised setting, where only a small proportion of target samples are labeled while most remain unlabeled. Based on this surprising observation, we formulate a novel problem termed *Semi-Supervised Noise Adaptation* (SSNA), which aims to leverage a synthetic noise domain to improve the generalization of the target domain. To address this problem, we first establish a generalization bound characterizing the effect of the noise domain on generalization, based on which we propose a Noise Adaptation Framework (NAF). Extensive experiments demonstrate that NAF effectively utilizes the noise domain to tighten the generalization bound of the target domain, thereby achieving improved performance. The codes are available at `https://anonymous.4open.science/r/SSNA`.

## 1 Introduction

Transfer Learning (TL) (Pan & Yang, 2010; Yang et al., 2020) aims to transfer knowledge from a label-rich source domain to a related but label-scarce target domain. Most TL approaches have been proposed (Pan & Yang, 2010; Day & Khoshgoftaar, 2017; Jiang et al., 2022; Yang et al., 2020; Bao et al., 2023), demonstrating substantial progress in various practical applications (Gu et al., 2022; Yao et al., 2019; Meegahapola et al., 2024; Ren et al., 2024). While the source and target domains often exhibit distributional divergence, the source domain typically contains semantically meaningful samples (*e.g.*, images, text, or audio) that provide a crucial foundation for effective knowledge transfer. However, a recent study (Yao et al., 2025) has made a surprising finding: *Noise drawn from simple distributions (e.g., Gaussian distributions), can also serve as a viable source domain, provided that its discriminability and transferability are preserved.* Although noise is generally viewed as semantically meaningless and even detrimental, empirical evidence has demonstrated that knowledge can be transferred from the noise domain to the target domain in the Semi-Supervised Learning (SSL) setting, where most target samples are unlabeled and only a small subset is labeled. This observation is particularly valuable, as concerns related to privacy, confidentiality, and copyright often hinder the acquisition of feasible source samples. However, this study has two key limitations: (i) it lacks a generalization bound analysis explaining why the noise domain improves generalization; and (ii) its experiments omit standard benchmark datasets such as CIFAR-10/100 (Krizhevsky et al., 2009) and ImageNet-1K (Deng et al., 2009), limiting the generalizability of its findings.

Motivated by those limitations, we formalize a novel problem termed Semi-Supervised Noise Adaptation (SSNA), as illustrated in Figure 1. Under the SSNA setting, we define a *target* domain and a *noise* domain. The target domain comprises a small proportion of labeled samples, with most remaining unlabeled. In contrast, the noise domain is generated from random distributions and serves as a surrogate source domain. *Since noise inherently lacks semantic meanings, we follow (Yao et al., 2025) and randomly and uniquely assign the class indices from the target domain to each noise class in a one-to-one manner* (see solid arrow in Figure 1). Accordingly, the learning tasks in both domains are aligned. The objective of SSNA is to enhance the generalization of the target domain by leveraging both labeled and unlabeled target samples, as well as noise.

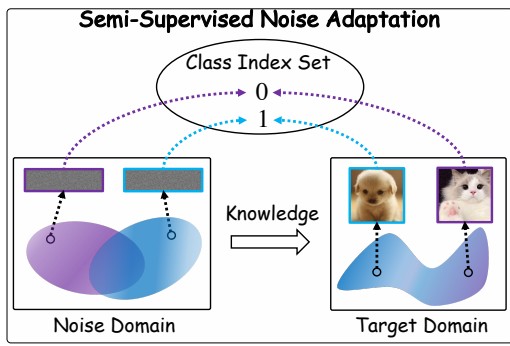 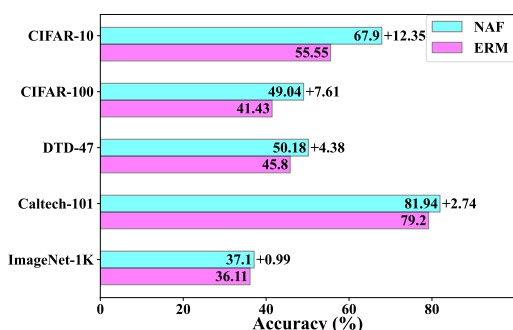

Figure 1: SSNA: The target domain includes a limited number of labeled samples, with most remaining unlabeled, while the noise domain is generated from random distributions. Noise classes, lacking semantic meaning, are mapped one-to-one to target classes (see solid arrows). The goal is to improve the generalization of the target domain by utilizing the noise domain.

Figure 2: Accuracy (%) of NAF and ERM on five benchmark datasets, *i.e.*, CIFAR-10, CIFAR-100, DTD-47, Caltech-101, and ImageNet-1K, using ResNet-18 (He et al., 2016). NAF consistently outperforms ERM across all the datasets, demonstrating the effectiveness of NAF in transferring knowledge from the noise domain to the target domain.

To address this problem, we first establish a generalization bound characterizing the effect of the noise domain on generalization. Based on this theoretical insight, we propose a Noise Adaptation Framework (NAF) that projects target samples and noise into a domain-invariant representation space by minimizing the empirical risks of both domains and reducing their distributional divergence. Optimizing NAF's objective effectively tightens the target domain's generalization bound, thereby improving its generalization performance. Experimental results on benchmark datasets demonstrate the effectiveness of NAF compared with Empirical Risk Minimization (ERM), a standard supervised learning baseline. As shown in Figure 2, NAF consistently outperforms ERM by up to 12.35%, 7.61%, 4.38%, and 2.74% on CIFAR-10, CIFAR-100, DTD-47, and Caltech-101, respectively, with 4 labeled samples per class. Moreover, on the more challenging ImageNet-1K dataset with 1000 classes and 100 labeled samples per class, NAF achieves an improvement of up to 0.99% over ERM.

The main contributions of this paper are summarized as follows. (1) We introduce the SSNA problem, providing a fresh perspective on the utilization of noise. (2) We provide a generalization bound of SSNA that characterizes the impact of the noise domain on generalization, based on which we propose the NAF. (3) Extensive experiments demonstrate that NAF can effectively tighten the generalization bound of the target domain, leading to better generalization performance.

## 2 RELATED WORK

Our work is closely related to TL (Pan & Yang, 2010; Yang et al., 2020) and semi-supervised learning (SSL) (Van Engelen & Hoos, 2020; Gui et al., 2024), ***both of which aim to leverage unlabeled samples to improve the generalization of the target domain***.

*TL enhances generalization by leveraging abundant labeled source samples to guide the learning of unlabeled target samples*. Ben-David et al. (2006; 2010) introduce the theoretical foundations for TL by establishing a generalization bound for the target domain. Based on this theoretical bound, a key objective in TL is to minimize the distributional discrepancy between the source and target domains. To this end, various distribution alignment methods have been proposed, primarily leveraging Maximum Mean Discrepancy (MMD) (Gretton et al., 2006) and Adversarial Domain Alignment (ADA) (Ganin et al., 2016). For instance, several studies (Long et al., 2013; 2015; 2019; Yao et al., 2019; Cheng et al., 2024) propose MMD variants to quantify the distributional divergence between the source and target domains. Another line of research (Ganin et al., 2016; Long et al., 2018; Liu et al., 2021; Gao et al., 2021; Shi & Liu, 2023; Meegahapola et al., 2024) explores diverse forms of ADA, which mitigate this divergence via a min-max game between a feature extractor and a domain discriminator. Furthermore, several studies (Gu et al., 2022; Bai et al., 2024; Liu et al., 2024; Ren et al., 2024) utilize other distributional alignment mechanisms to facilitate cross-domain

knowledge transfer. Note that most of the above studies, *the source domain consists of semantically meaningful samples* (*e.g.*, images, text, or audio).

*SSL utilizes a few labeled target samples to guide the learning of unlabeled target samples.* Many methods (Xie et al., 2020; Sohn et al., 2020; Zhang et al., 2021; Chen et al., 2022; Wang et al., 2022) utilize data augmentation and pseudo-label refinement mechanisms, where the former improves sample diversity and the latter mitigates pseudo-label bias. For instance, UDA (Xie et al., 2020) strengthens consistency training by replacing simple noise injection with strong data augmentation. FixMatch (Sohn et al., 2020) generates pseudo-labels from weakly augmented samples and enforces consistency with their strongly augmented counterparts. FlexMatch (Zhang et al., 2021) further refines this method by dynamically adjusting class-specific confidence thresholds. To alleviate pseudo-label bias, DST (Chen et al., 2022) decouples pseudo-label generation and utilization with two independent classifiers while adversarially optimizing the representation extractor. DebiasMatch (Wang et al., 2022) uses causal inference to adjust decision margins based on pseudo-label imbalance. Another line of research (Grandvalet & Bengio, 2004; Cui et al., 2020; Zhang et al., 2024) focuses on directly guiding the learning of unlabeled samples. A recent example is LERM (Zhang et al., 2024), which utilizes class-specific label-encodings to guide the learning of unlabeled samples.

Our work is primarily motivated by (Yao et al., 2025), which reveals that noise drawn from simple distributions (*e.g.*, Gaussian distributions) contains transferable knowledge, as long as its discriminability and transferability are preserved. This may initially appear counter-intuitive, as noise is typically viewed as semantically meaningless and potentially harmful. In practice, however, several studies (Baradad Jurjo et al., 2021; Li, 2022; Huang et al., 2025; Wang et al., 2025; Tang et al., 2022; Luo et al., 2021) have explored the potential of noise in addressing diverse machine learning tasks. For example, Baradad Jurjo et al. (2021) leverage noise to pre-train a visual representation model using a contrastive loss, resulting in better downstream performance. Another line of research (Huang et al., 2025; Wang et al., 2025) builds on the concept of *positive-incentive noise* introduced by (Li, 2022), leveraging it to augment original samples or representations, aiming to enhance generalization performance. Moreover, Luo et al. (2021); Tang et al. (2022) propose utilizing noise to tackle the distribution heterogeneity issue across clients in federated learning.

In summary, unlike the aforementioned studies, ***our work explores how the noise domain can be leveraged to facilitate the learning of unlabeled target samples in SSL within a TL framework***.

## 3 PROBLEM FORMULATION

In this section, we formulate the SSNA problem. Let $\mathcal{C} = \{0, \ldots, C-1\}$ be the class index set, where $C$ denotes the total number of classes. Let $\mathcal{E}$ and $\mathcal{X}$ denote the noise space (*e.g.*, a $p$-dimensional space) and the sample space (*e.g.*, a pixel-level image space), respectively.

**Definition 1.** *(Target Domain). The target domain is defined as $\mathcal{D}_t = \mathcal{D}_l \cup \mathcal{D}_u \cup \mathcal{D}_e$, where all samples lie in the sample space $\mathcal{X}$. Specifically, $\mathcal{D}_l = \{(\mathbf{x}_i^l, y_i^l)\}_{i=1}^{n_l}$ consists of labeled target samples, where each sample $\mathbf{x}_i^l$ is associated with a semantic class (e.g., "dog") that is mapped to an integer label $y_i^l \in \mathcal{C}$. $\mathcal{D}_u = \{\mathbf{x}_i^u\}_{i=1}^{n_u}$ and $\mathcal{D}_e = \{\mathbf{x}_i^e\}_{i=1}^{n_e}$ include the unlabeled and test target samples, respectively. Furthermore, the number of labeled target samples is much smaller than that of unlabeled target samples, i.e., $n_l \ll n_u$.*

**Definition 2.** *(Noise Domain). The noise domain is defined as $\mathcal{D}_n = \{(\mathbf{n}_i, y_i)\}_{i=1}^{n}$, where each noise $\mathbf{n}_i$ is drawn from a random distribution over $\mathcal{E}$. The corresponding label $y_i \in \mathcal{C}$ serves purely as an integer identifier without any semantic information.*

**Definition 3.** *(SSNA). Given a target domain $\mathcal{D}_t$, the objective of SSNA is to train a high-quality model $h_{\boldsymbol{\theta}*}$ using samples from $\mathcal{D}_l$, $\mathcal{D}_u$, and noise from $\mathcal{D}_n$, and then apply $h_{\boldsymbol{\theta}*}$ to classify the samples in $\mathcal{D}_e$ for evaluation.*

## 4 GENERALIZATION BOUND ANALYSIS AND EMPIRICAL VERIFICATION

In this section, we first present a generalization bound analysis for SSNA, from which NAF is derived and empirically shown to tighten the bound by leveraging the noise domain.

## 4.1 GENERALIZATION BOUND ANALYSIS

Before presenting the generalization bound for SSNA, we first address two fundamental questions based on the findings in (Yao et al., 2025):

> (i) *What knowledge is contained in the noise domain that can benefit the target domain?*

> (ii) *Is the semi-supervised setting in the target domain necessary?*

Regarding question (i), although the noise domain is constructed by random sampling from a noise space, it shares the same class index set with the target domain (see Figure 3), thereby aligning their learning tasks. Concretely, the target domain contains $C$ classes indexed by $\{0, \ldots, C-1\}$. Accordingly, we set the number of noise classes to $C$ and sample noise for each class from a distinct Gaussian distribution. All noise drawn from each Gaussian distribution is assigned a distinct class index in $\{0, \ldots, C-1\}$ prior to training, establishing a fixed one-to-one correspondence between noise and target classes. Classifying noise into distinct class indices induces a ***discriminative structure*** in the representation space, *i.e.*, ***noise with the same class index forms compact clusters, whereas those with different class indices are separated***. Although the noise domain itself lacks semantic meaning, this induced structure provides valuable knowledge for transfer. For example, in Figure 3, class "0" in the noise domain carries no semantics, yet it corresponds to "cat" in the target domain. During distribution alignment, noise from class "0" is aligned with "cat" representations, enforcing structural alignment across domains. Consequently, the discriminative structure of the noise domain serves as guidance, facilitating clearer class separation in the target domain.

As for question (ii), without labeled target samples to align the class indices between the noise and target domains, a classifier trained solely on the noise domain cannot effectively classify target samples. This is because the noise is randomly generated and does not originate from the same sample space as the target domain, lacking any inherent relationship with the target samples. Consequently, *a few labeled target samples are needed to bridge the two domains by aligning their class indices, enabling the effective transfer of discriminative structure from the noise domain to the target domain* (see **Q5** in Section 5.3 for a detailed analysis).

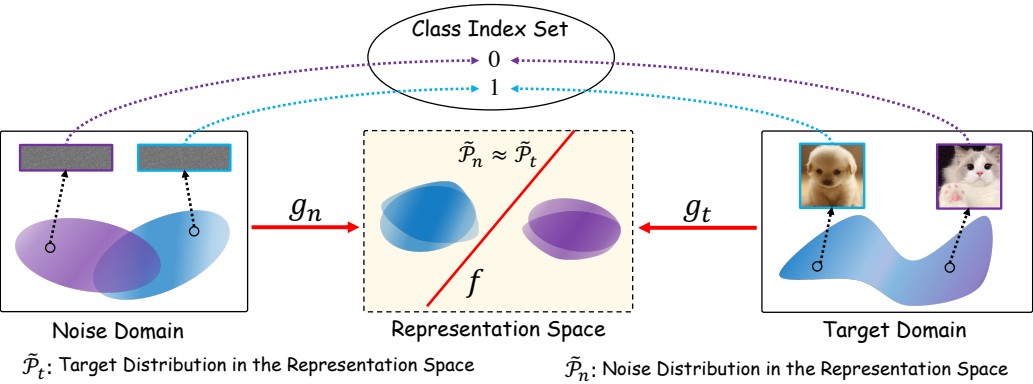

Figure 3: Under the SSNA setting, although the noise domain is generated from a random distribution, it shares a common set of class indices with the target domain. By classifying noise into distinct class indices in the representation space, a discriminative structure is formed that guides the alignment with the target domain and enhances the separability of target representations.

Next, we apply the theoretical framework of semi-supervised TL in (Ben-David et al., 2010) to analyze the generalization bound of SSNA. Since the noise does not originate from the same sample space as the target domain, it is infeasible to directly measure the distributional discrepancy between them. To address this issue, *we project both domains into a domain-shared representation space $\mathcal{Z}$ and derive the generalization bound for the target domain within this space.* Specifically, let $\mathcal{F}$ be a hypothesis space over $\mathcal{Z}$, consisting of functions $f : \mathcal{Z} \to \{0, 1\}$ with VC dimension $d$. Denote by $\widetilde{\mathcal{P}}_t$ and $\widetilde{\mathcal{P}}_n$ the target and noise distributions over $\mathcal{Z}$, respectively. Let $\mathcal{U}_t, \mathcal{U}_n$ be unlabeled samples of size $m'$ each, drawn *i.i.d.* from $\widetilde{\mathcal{P}}_t$ and $\widetilde{\mathcal{P}}_n$, respectively. Let $\mathcal{L}_t$ and $\mathcal{L}_n$ be labeled samples of sizes $\beta m$ and $(1-\beta)m$, drawn *i.i.d.* from $\widetilde{\mathcal{P}}_t$ and $\widetilde{\mathcal{P}}_n$, respectively. Define $\hat{\epsilon}_\alpha(f) = \alpha \hat{\epsilon}_t(f) + (1-\alpha)\hat{\epsilon}_n(f)$

($\alpha \in [0, 1]$) as the convex combination of the empirical target error $\hat{\epsilon}_t(f)$ and empirical noise error $\hat{\epsilon}_n(f)$, measured on $\mathcal{L}_t$ and $\mathcal{L}_n$, respectively. Based on those notations summarized in Table 9 of Appendix C.1, we present the generalization bound of SSNA in a two-domain setting in Theorem 1.

**Theorem 1.** *(Generalization Bound of SSNA) Let $\hat{f} = \arg\min_{f \in \mathcal{F}} \hat{\epsilon}_\alpha(f)$ be the empirical minimizer of $\hat{\epsilon}_\alpha(f)$, and let $f_t^* = \arg\min_{f \in \mathcal{F}} \epsilon_t(f)$ be the target error minimizer. Then, for any $\delta \in (0, 1)$, with probability at least $1 - \delta$ (over the choice of the samples), we have:*

$$\epsilon_t(\hat{f}) \le \epsilon_t(f_t^*) + \mathcal{O}\left(\gamma\sqrt{\frac{d\log m + \log(\frac{1}{\delta})}{m}}\right) + 2(1-\alpha)\left[\frac{1}{2}\hat{d}_{\mathcal{H}\Delta\mathcal{H}}(\mathcal{U}_n, \mathcal{U}_t) + \mathcal{O}\left(\sqrt{\frac{d\log m' + \log(\frac{1}{\delta})}{m'}}\right)\right.$$

$$\left. + \hat{\epsilon}_n(\hat{f}) + \hat{\epsilon}_t(\hat{f}) + \mathcal{O}\left(\sqrt{\frac{d\log(\frac{(1-\beta)m}{d}) + \log(\frac{1}{\delta})}{(1-\beta)m}}\right) + \mathcal{O}\left(\sqrt{\frac{d\log(\frac{\beta m}{d}) + \log(\frac{1}{\delta})}{\beta m}}\right)\right],$$

*where $\gamma = \sqrt{\frac{\alpha^2}{\beta} + \frac{(1-\alpha)^2}{1-\beta}}$, and $\hat{d}_{\mathcal{H}\Delta\mathcal{H}}(\mathcal{U}_n, \mathcal{U}_t)$ is the empirical $\mathcal{H}$-divergence estimated from noise and target samples in $\mathcal{Z}$.*

The proof is provided in Appendix C.2. Theorem 1 builds upon Theorem 3 in (Ben-David et al., 2010) and incorporates key insights from (Li et al., 2021). The resulting bound explicitly accounts for three key terms: (i) the empirical noise error $\hat{\epsilon}_n(\hat{f})$; (ii) the empirical target error $\hat{\epsilon}_t(\hat{f})$; and (iii) the empirical distributional discrepancy $\hat{d}_{\mathcal{H}\Delta\mathcal{H}}(\mathcal{U}_n, \mathcal{U}_t)$, without involving the joint optimal error term $\lambda$. Theorem 1 suggests that, *regardless of the origin of the source domain (e.g., images, text, or synthetic noise), the generalization bound on the expected target error can be tightened when those three terms are effectively reduced in $\mathcal{Z}$.* Moreover, it relaxes the common semi-supervised TL assumption that source and target domains must be related, explaining why even a synthetic noise domain can serve as an effective surrogate. Next, we empirically verify this theoretical insight.

## 4.2 EMPIRICAL VERIFICATION OF THEOREM 1

To empirically verify Theorem 1, we first present the proposed NAF based on this theorem, and then report several key results.

Building on Theorem 1, the generalization bound on the expected target error $\epsilon_t(\hat{f})$ can be minimized by jointly reducing $\hat{\epsilon}_t(\hat{f})$, $\hat{\epsilon}_n(\hat{f})$, and $\hat{d}_{\mathcal{H}\Delta\mathcal{H}}(\mathcal{U}_n, \mathcal{U}_t)$ in $\mathcal{Z}$. Accordingly, we design NAF to project target samples and noise into $\mathcal{Z}$ by minimizing three components: (i) $\mathcal{L}_t$: the empirical risk of labeled target samples, corresponding to $\hat{\epsilon}_t(\hat{f})$; (ii) $\mathcal{L}_n$: the empirical risk of noise, corresponding to $\hat{\epsilon}_n(\hat{f})$; and (iii) $\mathcal{L}_{n,t}$: the distributional discrepancy between projected domains, whose minimization implicitly reduces $\hat{d}_{\mathcal{H}\Delta\mathcal{H}}(\mathcal{U}_n, \mathcal{U}_t)$. Thus, the optimization objective of the NAF is formulated by

$$\min_{g_t, g_n, f} \mathcal{L}_t(\mathcal{D}_l; g_t, f) + \alpha\mathcal{L}_n(\mathcal{D}_n; g_n, f) + \beta\mathcal{L}_{n,t}(\mathcal{D}_l, \mathcal{D}_u, \mathcal{D}_n; g_t, g_n, f), \quad (1)$$

where $g_t(\cdot)$ is a representation extractor projecting target samples from $\mathcal{X}$ to $\mathcal{Z}$, $g_n(\cdot)$ is a noise projector mapping noise from $\mathcal{E}$ to $\mathcal{Z}$, $f(\cdot)$ is a classifier (see Figure 3), and $\alpha, \beta$ are two positive trade-off parameters to control the importance of $\mathcal{L}_n$ and $\mathcal{L}_{n,t}$, respectively. By optimizing the problem (1), the generalization bound of the target domain can be effectively tightened, thereby improving the generalization performance.

NAF is formulated as a general framework with flexible instantiations for its components. In the implementation, $\mathcal{L}_t$ and $\mathcal{L}_n$ are instantiated with the *cross-entropy loss*, and $\mathcal{L}_{n,t}$ can be realized through various distribution alignment mechanisms. In practice, we design five mechanisms and empirically adopt the *Negative Domain Similarity* (NDS) mechanism, while detailed analyses of alternative designs are provided in **Q7** of Section 5.3. NDS measures the discrepancy between the projected target and noise domains by computing the cosine similarities between their global means and class-wise means, averaging those similarities, and then negating the result (see details in Appendix A). Moreover, we use the classifier $f(\cdot)$ to assign pseudo-labels to unlabeled target samples and iteratively update them to estimate class means.

Next, we present empirical results showing that NAF achieves a tighter generalization bound on the target domain compared to the supervised learning baseline, *i.e.*, ERM, which uses only $\mathcal{L}_t$. To

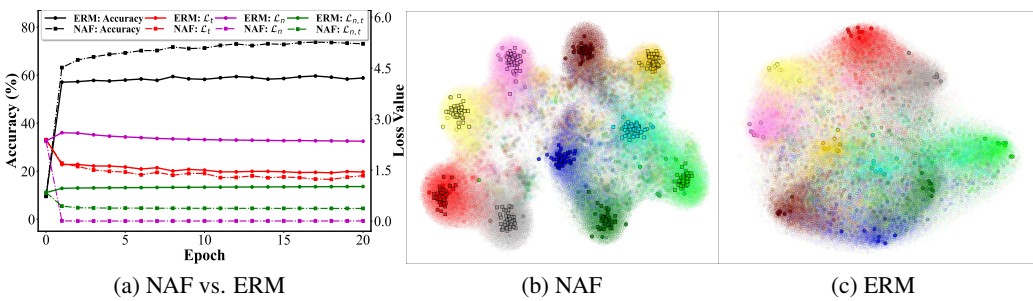

Figure 4: (a) Training loss and accuracy curves for NAF and ERM on CIFAR-10 with ResNet-18. $\mathcal{L}_t$ denotes the empirical risk of labeled target samples, $\mathcal{L}_n$ is the empirical risk of noise, and $\mathcal{L}_{n,t}$ measures the distributional discrepancy between domains. (b) Representations learned by NAF on CIFAR-10 with ResNet-18, where '■' indicates noise representation; '●' and '○' represent labeled and unlabeled target representations, respectively. (c) Representations learned by ERM on CIFAR-10 with ResNet-18, with the same symbol scheme as in (b). Colors correspond to different classes.

construct a noise domain, we first sample $C$ class means from a standard Gaussian distribution in a 1024-dimensional space. For each class, we then assign an identity covariance matrix. Based on each class mean and its corresponding covariance matrix, we then sample 50 noise from the associated Gaussian distribution to form the noise domain. Figure 4a plots the training trajectories of $\mathcal{L}_t$, $\mathcal{L}_n$, and $\mathcal{L}_{n,t}$, along with the test accuracy curves for NAF and ERM on CIFAR-10 using ResNet-18, with 4 labeled samples per class. Several insightful observations can be drawn.

- Both methods demonstrate notable reductions in $\mathcal{L}_t$, as it is explicitly minimized in their respective objective functions.

- The values of $\mathcal{L}_n$ and $\mathcal{L}_{n,t}$ in ERM are consistently higher than those in NAF, which is reasonable since ERM does not explicitly minimize them.

- When $\mathcal{L}_t$ is jointly minimized with $\mathcal{L}_n$ and $\mathcal{L}_{n,t}$ in NAF, the accuracy significantly improves over ERM. *Since $\mathcal{L}_n$ and $\mathcal{L}_{n,t}$ are derived from the noise domain, this improvement indicates that incorporating the noise domain tightens the target generalization bound, producing positive transfer*. This observation aligns with the theoretical result in Theorem 1.

Furthermore, we visualize the representations learned by NAF and ERM in the above experiment using t-SNE (Van der Maaten & Hinton, 2008). As shown in Figure 4b, NAF produces a clear discriminative structure, where noise representations from different classes form well-separated clusters and align closely with the corresponding target representations. Notably, because the noise is fed into $g_n(\cdot)$ solely to generate its representations, the discriminative structure observed in the noise domain arises from the predefined noise distributions and from the supervised training applied to noise representations in the representation space. In contrast, ERM, as plotted in Figure 4c, exhibits less discriminable target representations. This difference can be attributed to the joint minimization of $\mathcal{L}_n$ and $\mathcal{L}_{n,t}$: *minimizing $\mathcal{L}_n$ enforces noise representations to form compact and well-separated clusters across classes, and minimizing $\mathcal{L}_{n,t}$ aligns all target representations with those clusters, thus producing more discriminative target representations*.

## 5 EXPERIMENTS

### 5.1 SETUP

**Datasets**. We use the following benchmark datasets: CIFAR-10 (Krizhevsky et al., 2009), CIFAR-100 (Krizhevsky et al., 2009), DTD (Cimpoi et al., 2014), Caltech-101 (Fei-Fei et al., 2004), CUB-200-2011 (Wah et al., 2011), Oxford Flowers-102 (Nilsback & Zisserman, 2008), Stanford Cars-196 (Krause et al., 2013), ImageNet-1K (Deng et al., 2009), and AG News-4 (Zhang et al., 2015). For the first seven vision datasets, we randomly select four labeled samples per class from the original training set, treating the remaining samples as unlabeled; for ImageNet-1K, we sample 100 labeled examples per class due to its large scale, with the rest used as unlabeled data. AG News-4 is a text

classification dataset consisting of news articles from four categories, for which we randomly draw four labeled samples and 1,000 unlabeled samples. Further details are provided in Appendix B.1.

**Noise Domain Construction**. For consistency and simplicity across tasks, we construct the noise domain using the produce described in Section 4.2, unless otherwise stated.

**Evaluation Metric.** We evaluate performance using the classification accuracy in $\mathcal{D}_e$. For a fair comparison, we report the accuracy of the last epoch. In most cases, results are averaged over three independent runs, while single-run accuracy is reported in certain settings (*e.g.*, ImageNet-1K).

## 5.2 MAIN EXPERIMENTS

**Q1. How does NAF perform compared to ERM on standard classification benchmarks?** Table 1 lists the results on CIFAR-10, CIFAR-100, DTD-47 and Caltech-101 using ResNet-18 and ResNet-50. As shown, NAF consistently outperforms ERM, which represents the standard supervised baseline, across all datasets. In particular, NAF yields notable Top-1 accuracy improvements of 12.35% and 15.15% over ERM on CIFAR-10 with ResNet-18 and ResNet-50, respectively. This consistent advantage over ERM confirms that NAF achieves positive transfer from the noise domain to the target domain. The reason is that NAF introduces the noise domain with class-discriminative structure and enforces distributional alignment between the noise and target domains. This process encourages all target representations to form more separable clusters, which enhances class discriminability and thereby improves the generalization of the target domain.

Table 1: Accuracy (%) comparison on CIFAR-10 and CIFAR-100, DTD-47, and Caltech-101 using ResNet-18 and ResNet-50, respectively. Here, $\Delta$ indicates the performance gain introduced by NAF.

| Datasets | CIFAR-10 | | CIFAR-100 | | DTD-47 | | Caltech-101 | |
|---|---|---|---|---|---|---|---|---|
| ResNet-18 | Top-1 | Top-5 | Top-1 | Top-5 | Top-1 | Top-5 | Top-1 | Top-5 |
| ERM | 55.55 | 92.85 | 41.43 | 71.40 | 45.80 | 74.26 | 79.20 | 93.29 |
| NAF | 67.90 | 96.38 | 49.04 | 80.56 | 50.18 | 77.98 | 81.94 | 95.01 |
| $\Delta$ | **+12.35** | **+3.53** | **+7.61** | **+9.16** | **+4.38** | **+3.72** | **+2.74** | **+1.72** |
| ResNet-50 | Top-1 | Top-5 | Top-1 | Top-5 | Top-1 | Top-5 | Top-1 | Top-5 |
| ERM | 58.83 | 94.25 | 46.71 | 76.53 | 49.56 | 76.65 | 81.99 | 94.70 |
| NAF | 73.98 | 97.01 | 52.82 | 82.16 | 53.97 | 79.68 | 84.41 | 96.14 |
| $\Delta$ | **+15.15** | **+2.76** | **+6.11** | **+5.63** | **+4.41** | **+3.03** | **+2.42** | **+1.44** |

**Q2. Can NAF achieve improvements over ERM on fine-grained classification tasks?** Table 2 presents the results on three fine-grained classification datasets, including CUB-200, OxfordFlowers-102, and StanfordCars-196, using ResNet-18. As observed, NAF consistently outperforms ERM by a large margin across all datasets. Those results demonstrate that NAF can effectively leverage the noise domain to achieve positive transfer in fine-grained classification tasks.

Table 2: Accuracy (%) comparison on fine-grained classification datasets using ResNet-18.

| Datasets | CUB-200 | OxfordFlowers-102 | StanfordCars-196 |
|---|---|---|---|
| ERM | 41.92 | 81.07 | 28.01 |
| NAF | 50.86 | 86.58 | 35.75 |
| $\Delta$ | **+8.94** | **+5.51** | **+7.74** |

**Q3. Does NAF scale to large-scale datasets such as ImageNet?** We evaluate NAF on TinyImageNet-200 and ImageNet-1K with 100 labeled samples per class using ResNet-18 to assess its performance on medium- and large-scale datasets. NAF achieves an accuracy of 37.10%, outperforming ERM (36.11%) by 0.99%. This result further highlights NAF's effectiveness, even on large-scale datasets with 1,000 classes, demonstrating its potential for addressing complex real-world challenges.

**Q4. Is NAF effective on text categorization tasks?** To assess the applicability of NAF beyond visual classification, we conduct experiments on AG News-4 (Zhang et al., 2015). Here, texts are encoded using a pre-trained BERT model, and noise is mapped through a nonlinear projector with ReLU activation. NAF achieves an accuracy of 82.82%, outperforming ERM, which achieves 78.64%. The results suggest that NAF could potentially facilitate knowledge transfer in non-visual tasks.

**Q5. Is NAF effective as a plug-in when combined with existing SSL methods?** To investigate this question, we conduct experiments using six state-of-the-art (SOTA) SSL methods: UDA (Xie et al.,

2020), FixMatch (Sohn et al., 2020), FlexMatch (Zhang et al., 2021), DebiasMatch (Wang et al., 2022), DST (Chen et al., 2022), and LERM (Zhang et al., 2024). NAF can be seamlessly integrated as a plugin into those SOTA SSL methods by incorporating $\mathcal{L}_n$ and $\mathcal{L}_{n,t}$ into their objective functions. Table 3 reports the results at the 5th, 10th, 15th, and 20th epochs on CIFAR-10 and CIFAR-100 using ResNet-18. We observe that incorporating NAF leads to consistent performance gains across all SSL methods. Specifically, NAF improves accuracy by 20.83% and 9.91% over UDA and FixMatch, respectively, at the 20th epoch on CIFAR-10. Those results indicate that NAF effectively enhances the generalization of SOTA methods by transferring knowledge from the noise domain. Additional results on DTD-47 and Caltech-101 are offered in Appendix D.

Table 3: Accuracy (%) comparison on CIFAR-10 and CIFAR-100 using ResNet-18. Here, $\Delta$ indicates the performance gain introduced by NAF.

| Datasets | CIFAR-10 | | | | | CIFAR-100 | | | | |
|---|---|---|---|---|---|---|---|---|---|---|
| Epoch | 5 | 10 | 15 | 20 | Average | 5 | 10 | 15 | 20 | Average |
| UDA (Xie et al., 2020) | 51.67 | 55.37 | 56.03 | 56.11 | 54.80 | 38.30 | 42.99 | 45.93 | 47.41 | 43.66 |
| UDA + NAF | 73.55 | 76.16 | 76.52 | 76.94 | 75.79 | 40.37 | 45.44 | 47.82 | 48.80 | 45.61 |
| $\Delta$ | +21.88 | +20.79 | +20.49 | +20.83 | +20.99 | +2.07 | +2.45 | +1.89 | +1.39 | +1.95 |
| FixMatch (Sohn et al., 2020) | 66.41 | 68.41 | 69.01 | 69.40 | 68.31 | 39.38 | 40.78 | 41.98 | 42.45 | 41.15 |
| FixMatch + NAF | 75.51 | 77.89 | 79.00 | 79.31 | 77.93 | 40.97 | 43.28 | 44.06 | 44.93 | 43.31 |
| $\Delta$ | +9.10 | +9.48 | +9.99 | +9.91 | +9.62 | +1.59 | +2.50 | +2.08 | +2.48 | +2.16 |
| FlexMatch (Zhang et al., 2021) | 73.61 | 79.85 | 83.46 | 84.53 | 80.36 | 45.41 | 50.28 | 51.91 | 54.30 | 50.48 |
| FlexMatch + NAF | 79.22 | 82.72 | 84.32 | 84.90 | 82.79 | 48.10 | 52.91 | 54.97 | 55.73 | 52.93 |
| $\Delta$ | +5.61 | +2.87 | +0.86 | +0.37 | +2.43 | +2.69 | +2.63 | +3.06 | +1.43 | +2.45 |
| DebiasMatch (Wang et al., 2022) | 68.71 | 77.68 | 79.86 | 82.04 | 77.07 | 46.71 | 51.97 | 54.73 | 56.30 | 52.43 |
| DebiasMatch + NAF | 76.12 | 80.89 | 82.54 | 83.05 | 80.65 | 49.57 | 54.02 | 56.36 | 57.45 | 54.35 |
| $\Delta$ | +7.41 | +3.21 | +2.68 | +1.01 | +3.58 | +2.86 | +2.05 | +1.63 | +1.15 | +1.92 |
| DST (Chen et al., 2022) | 78.40 | 82.84 | 84.48 | 85.47 | 82.80 | 45.40 | 49.74 | 51.68 | 53.17 | 50.00 |
| DST + NAF | 80.70 | 83.46 | 84.87 | 85.53 | 83.64 | 48.73 | 52.28 | 54.10 | 54.93 | 52.51 |
| $\Delta$ | +2.30 | +0.62 | +0.39 | +0.06 | +0.84 | +3.33 | +2.54 | +2.42 | +1.76 | +2.51 |
| LERM (Zhang et al., 2024) | 60.03 | 62.42 | 63.81 | 64.77 | 62.76 | 48.10 | 50.13 | 50.83 | 51.66 | 50.18 |
| LERM + NAF | 66.01 | 67.34 | 67.83 | 68.00 | 67.30 | 49.42 | 51.06 | 51.65 | 51.97 | 51.03 |
| $\Delta$ | +5.98 | +4.92 | +4.02 | +3.23 | +4.54 | +1.32 | +0.93 | +0.82 | +0.31 | +0.85 |

## 5.3 ANALYSIS

**Q6. How does the impact of NAF change as the number of labeled target samples varies?** Table 4 reports the results on CIFAR-10 using ResNet-18 with different numbers of labeled samples per class. We have several insightful observations. (1) When the number of labeled target samples is zero, both ERM and NAF perform poorly. For ERM, the absence of labeled target samples hinders the effective learning of unlabeled samples, resulting in significant performance degradation. In NAF, the noise comes from a space different from that of the target domain, and the target samples are unlabeled. As a result, the class-discriminative structure of the noise cannot be effectively aligned with the target domain. (2) When the number of labeled target samples is non-zero, NAF outperforms ERM across all scenarios. Those results indicate that NAF effectively leverages both labeled target samples and noise to guide the learning of unlabeled target samples, enhancing the generalization of the target domain.

Table 4: Accuracy (%) comparison on CIFAR-100 using ResNet-18 with different numbers of labeled target samples per class.

| # Labeled target samples per class | 0 | 4 | 8 | 12 | 16 | 20 |
|---|---|---|---|---|---|---|
| ERM | 0.97 | 42.24 | 54.11 | 58.27 | 61.64 | 63.85 |
| NAF | 1.34 | 49.98 | 59.51 | 62.21 | 64.23 | 66.45 |

**Q7. How do $\mathcal{L}_n$ and $\mathcal{L}_{n,t}$ influence the performance of NAF?** We examine two NAF variants: (1) NAF (w/o $\mathcal{L}_n$), which ablates $\mathcal{L}_n$; and (2) NAF (w/o $\mathcal{L}_{n,t}$), which removes $\mathcal{L}_{n,t}$. Additionally, ERM can be seen as a NAF variant that eliminates both $\mathcal{L}_n$ and $\mathcal{L}_{n,t}$. The results on CIFAR-100 using ResNet-18 are shown in Table 5. We observe that NAF outperforms all variants, indicating that both losses are beneficial. Moreover, NAF (w/o $\mathcal{L}_n$) outperforms NAF (w/o $\mathcal{L}_{n,t}$), suggesting that reducing distributional divergence between domains is more crucial.

Table 5: Accuracy (%) of NAF variants on CIFAR-100 using ResNet-18.

| ERM | NAF (w/o $\mathcal{L}_n$) | NAF (w/o $\mathcal{L}_{n,t}$) | NAF |
|---|---|---|---|
| 42.24 | 47.33 | 40.64 | 49.98 |

**Q8. How does NAF perform under different distribution alignment mechanisms?** NAF is a general framework that can incorporate various distribution alignment mechanisms, with NDS employed in our implementation. To verify the generality of NAF, we consider several alternative alignment strategies: (1) *Negative Sample Similarity* (NSS): It calculates the negative average cosine similarities between all noise-target pairs from the same class. (2) *Negative Contrastive Domain Similarity* (NCDS): It computes a contrastive loss (Radford et al., 2021) over class-wise means across the noise and target domains. (3) *Negative Contrastive Sample Similarity* (NCSS): It defines a regression loss that aligns the cosine similarity of each noise–target pair to a target value: $+1$ for same-class pairs and $-1$ for different-class pairs. (4) *Euclidean Domain Distance* (EDD): It computes the average Euclidean distance between the global and class-wise means of the noise and target domains. Their specific formulations are defined in Appendix A. Table 6 lists the results on CIFAR-100 using ResNet-18. NAF (NDS) achieves the highest performance, verifying that NDS effectively captures distributional divergence across domains. In contrast, NAF (EDD) performs the worst, suggesting that Euclidean distance may be less suitable than cosine-based measures in this context. NAF (NSS), NAF (NCDS), and NAF (NCSS) also outperform ERM, confirming the generality of NAF in accommodating different alignment mechanisms.

Table 6: Accuracy (%) of NAF with various distributional alignment mechanisms on CIFAR-100 using ResNet-18.

| NAF (NDS) | NAF (NSS) | NAF (NCDS) | NAF (NCSS) | NAF (EDD) | ERM |
|---|---|---|---|---|---|
| 49.98 | 48.65 | 47.20 | 44.27 | 20.03 | 42.24 |

**Q9. What happens when the noise domain loses its discriminative structure?** To verify the role of the discriminative structure of the noise domain, we evaluate a variant of NAF termed NAF with Single Point, *i.e.*, NAF (SP). In NAF (SP), a single noise vector is sampled from a standard Gaussian distribution and assigned to all classes, with each class receiving 50 identical copies, effectively removing any class-discriminative structure. On CIFAR-10, NAF (SP) achieves 33.34% accuracy, substantially lower than ERM's 58.15%. On CIFAR-100, the gap is even larger, with NAF (SP) at 6.79% versus ERM at 42.24%. The dramatic performance drop indicates that collapsing all noise to a single point causes negative transfer, as the noise domain no longer provides class-discriminative structure for domain alignment. This suggests that NAF leverages the class-discriminative structure in the noise domain to facilitate better generalization in the target domain, highlighting the importance of preserving class-discriminative structure in the noise domain.

**Q10. How does NAF perform under distinct noise generation strategies?** We conduct experiments by varying the noise generation strategies across three dimensions. (1) **Covariance Scale**: In the original setup, we first sample a mean for each class from a standard Gaussian distribution. Next, for each class, we generate individual noise from a Gaussian distribution with the corresponding mean and identity covariance $\mathbf{I}$. We additionally evaluate two configurations in which all class covariances are scaled to $0.1 \cdot \mathbf{I}$ and $10 \cdot \mathbf{I}$. (2) **Noise Dimensionality**: In the original setup, the noise dimensionality is set to 1024. We additionally evaluate two configurations with noise dimensionalities of 512 and 2048. (3) **Distribution Type**: In the original setup, the noise is drawn from a Gaussian distribution. We additionally test the log-normal distribution and the Laplace distribution. The results, listed in Table 7, indicate that NAF achieves comparable performance across a variety of noise settings, including variations in covariance scale, noise dimensionality, and distribution type. Those observations suggest that NAF can accommodate different noise configurations, highlighting its potential flexibility.

**Q11. Is NAF still effective when the target domain exhibits class imbalance?** We conduct an experiment on CIFAR-10 using ResNet-18 with a long-tailed setup. In this configuration, the labeled and unlabeled sets have per-class sample counts of [50, 30, 20, 10, 6, 4, 3, 2, 2, 1] and [1000, 600, 200, 100, 60, 40, 20, 10, 6, 4], respectively. NAF achieves an accuracy of 56.38% and a macro F1-score of 53.22%, outperforming ERM, which attains 51.19% accuracy and a macro F1-score of 45.73%. The results suggest that NAF remains effective even under such a class imbalance scenario.

Table 7: Accuracy (%) comparison on CIFAR-100 using ResNet-18 with noise drawn from various noise generation strategies. Here, $\boldsymbol{\mu}_c$ is the class mean belonging to class $c$, and $d$ is the dimensionality of the noise.

| Noise Configuration | Noise Distribution | Accuracy |
|---|---|---|
| Baseline | Gaussian: $\mathcal{N}(\boldsymbol{\mu}_c, \mathbf{I}), d = 1024$ | 49.98 |
| Covariance Scale | Gaussian: $\mathcal{N}(\boldsymbol{\mu}_c, 0.1 \cdot \mathbf{I}), d = 1024$ | 50.38 |
| | Gaussian: $\mathcal{N}(\boldsymbol{\mu}_c, 10 \cdot \mathbf{I}), d = 1024$ | 47.64 |
| Noise Dimensionality | Gaussian: $\mathcal{N}(\boldsymbol{\mu}_c, \mathbf{I}), d = 512$ | 49.44 |
| | Gaussian: $\mathcal{N}(\boldsymbol{\mu}_c, \mathbf{I}), d = 2048$ | 51.04 |
| Distribution Type | Log-normal: $\log \mathcal{N}(\boldsymbol{\mu}_c, \mathbf{I}), d = 1024$ | 48.31 |
| | Laplace: $\mathcal{L}((\boldsymbol{\mu}_c)_d, 1/\sqrt{2}), d = 1024$ | 49.99 |

**Q12. Is there another method to learn the noise domain in the representation space?** In the above experiments, we use a noise projector $g_n$ to learn an optimal noise domain in the representation space. As an alternative, we explore constructing an optimal noise domain by learning its mean $\mu$ and standard deviation $\sigma$, and apply the reparameterization trick (Kingma & Welling, 2014) to map samples from a standard normal distribution to a Gaussian distribution $\mathcal{N}(\mu, \sigma^2 \mathbf{I})$ in the representation space. We evaluate this method on CIFAR-10 using ResNet-18, achieving an accuracy of 70.60%, which is comparable to the performance of NAF of 71.83%, and exceeds ERM by 12.45%. Those results suggest that modeling a parametric noise distribution via the reparameterization trick is also a feasible and effective strategy.

We conduct additional analyses in Appendix E: (1) the effectiveness of using noise as a surrogate source domain compared to real samples; (2) the influence of the amount of noise; (3) the analysis of hyperparameter sensitivity; (4) the impact of constructing the noise domain solely with class means; (5) the effect of inter-class distances within the noise domain; (6) NAF vs. plug-in SSL modules; and (7) NAF vs. contrastive learning methods. Those analyses provide a deeper understanding of the underlying principles of NAF and further validate its effectiveness.

## 6 DISCUSSION

While SSNA introduces additional noise, it fundamentally differs from data augmentation. Data augmentation typically enriches the target distribution via interpolation (*e.g.*, mixup (Zhang et al., 2018)), transformations (*e.g.*, rotations (Zhang et al., 2021)), or generative models (*e.g.*, diffusion (Ho et al., 2020)). In contrast, SSNA first generates noise from simple distributions (e.g., Gaussian distributions), which may differ substantially from the target distribution. The noise and target domains are then aligned in a shared representation space, allowing the discriminative structure of the noise domain to guide the learning of target representations. Hence, SSNA is a domain-level adaptation problem rather than a data-level augmentation problem.

## 7 CONCLUSION

In this paper, we formulate the SSNA problem, which leverages a synthetic noise domain to facilitate the learning task in the target domain. To address this problem, we first derive a generalization bound for the target domain that offers a theoretical understanding of how incorporating a noise domain can influence generalization performance. Building on this bound, we propose the NAF, which jointly minimizes the empirical risks on both the noise and target domains while reducing their distributional divergence within a domain-shared representation space. Extensive experiments demonstrate that NAF effectively tightens the generalization bound of the target domain, resulting in improved performance. Our work explores the use of synthetic noise domains as surrogate source domains to enhance the generalization of the target domain. A promising direction for future work is to extend SSNA to broader real-world scenarios.

ETHICS STATEMENT

This work does not involve human subjects, sensitive data, or any applications that may pose ethical risks. The datasets used are publicly available and widely adopted in the research community. Our contributions lie in formulating the SSNA problem and developing the NAF to address it, without raising concerns related to privacy, fairness, security, or other ethical issues.

REPRODUCIBILITY STATEMENT

We are committed to ensuring the reproducibility of our work. To this end, we make our source code available at `https://anonymous.4open.science/r/SSNA` . The implementation details, including datasets, model architectures, and hyperparameters, are described in Section 5.1 and Appendix B. With the released code and documentation, all reported results can be readily reproduced by the research community.

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

The appendices provide additional details and results, covering the following contents.

- Appendix A: Mathematical details of distribution alignment mechanisms.
- Appendix B: Additional experimental settings.
- Appendix C: Notation and Proof of Theorem 1.
- Appendix D: Supplementary experimental results.
- Appendix E: Additional analysis experiments.
- Appendix F: Declaration of use of large language models.

## A  MATHEMATICAL DETAILS OF DISTRIBUTION ALIGNMENT MECHANISMS

NAF is a general framework that supports various instantiations of the loss term $\mathcal{L}_{n,t}$. In this paper, we consider five distinct instantiations: (1) Negative Domain Similarity (NDS), (2) Negative Sample Similarity (NSS), (3) Negative Contrastive Domain Similarity (NCDS), (4) Negative Contrastive Sample Similarity (NCSS), and (5) Euclidean Domain Distance (EDD). Their specific formulations are defined below.

(1) NDS computes the cosine similarities between their global means and class-wise means, averages those similarities, and then negates the result, defined by

$$\mathcal{L}_{n,t}^{\text{NDS}} = -\frac{1}{C+1} \sum_{c=0}^{C} \langle \widetilde{\mathbf{m}}_n^c, \widetilde{\mathbf{m}}_t^c \rangle, \tag{2}$$

where $C$ is the number of classes, and $\langle \cdot, \cdot \rangle$ denotes the inner product. The case $c = 0$ corresponds to the global mean calculated across all classes. $\widetilde{\mathbf{m}}_n^c$ and $\widetilde{\mathbf{m}}_t^c$ denote the $l_2$-normalized class-wise means of the noise and target domains for class $c$, respectively. $\widetilde{\mathbf{m}}_t^c$ is calculated using both labeled and unlabeled target samples, with class assignments for unlabeled samples inferred via hard pseudo-labels predicted by the classifier and iteratively updated during training. *This pseudo-labeling strategy is consistently applied across all mechanisms.*

(2) NSS calculates the negative average cosine similarities between all noise-target pairs from the same class:

$$\mathcal{L}_{n,t}^{\text{NSS}} = -\frac{1}{\sum_{c=1}^{C} n_c n_{t,c}} \sum_{c=1}^{C} \sum_{i=1}^{n_c} \sum_{j=1}^{n_{t,c}} \langle \widetilde{\mathbf{n}}_{i,c}, \widetilde{\mathbf{x}}_{j,c}^t \rangle, \tag{3}$$

where $n_c$ and $n_{t,c}$ denote the numbers of noise and target samples in class $c$, and $\widetilde{\mathbf{n}}_{i,c}$ and $\widetilde{\mathbf{x}}_{j,c}^t$ are the $l_2$-normalized representations of the $i$-th noise and $j$-th target samples in class $c$.

(3) NCDS computes a contrastive loss over class-wise means across the noise and target domains, which is formulated as

$$\mathcal{L}_{n,t}^{\text{NCDS}} = -\frac{1}{2C} \sum_{c=1}^{C} \left[ \ln \frac{\exp\left(\langle \widetilde{\mathbf{m}}_n^c, \widetilde{\mathbf{m}}_t^c \rangle\right)}{\sum_{c'=1}^{C} \exp\left(\langle \widetilde{\mathbf{m}}_n^c, \widetilde{\mathbf{m}}_t^{c'} \rangle\right)} + \ln \frac{\exp\left(\langle \widetilde{\mathbf{m}}_t^c, \widetilde{\mathbf{m}}_n^c \rangle\right)}{\sum_{c'=1}^{C} \exp\left(\langle \widetilde{\mathbf{m}}_t^c, \widetilde{\mathbf{m}}_n^{c'} \rangle\right)} \right]. \tag{4}$$

(4) NCSS defines a regression loss that aligns the cosine similarity of each noise–target pair to a target value: $+1$ for same-class pairs and $-1$ for different-class pairs:

$$\mathcal{L}_{n,t}^{\text{NCSS}} = \frac{1}{C} \left[ \frac{1}{n_t n} \sum_{j=1}^{n_t} \sum_{i=1}^{n} \left( \langle \widetilde{\mathbf{x}}_j^t, \widetilde{\mathbf{n}}_i \rangle - y_{j,i} \right)^2 \right], \tag{5}$$

where $n$ and $n_t$ denote the numbers of noise and target samples, respectively. $\widetilde{\mathbf{n}}_i$ and $\widetilde{\mathbf{x}}_j^t$ represent the $l_2$-normalized representations of the $i$-th noise and $j$-th target samples. $y_{i,j}$ is set to 1 if the two samples share the same class, and $-1$ otherwise.

(5) EDD computes the average Euclidean distance between the global and class-wise means of the noise and target domains, defined as

$$\mathcal{L}_{n,t}^{\text{EDD}} = \frac{1}{C+1} \sum_{c=0}^{C} \| \mathbf{m}_n^c - \mathbf{m}_t^c \|_2. \tag{6}$$

## B ADDITIONAL EXPERIMENTAL SETTINGS

### B.1 DATASET DETAILS

In the experiments, we adopt the following datasets:

- **CIFAR-10** (Krizhevsky et al., 2009): 60,000 natural images across 10 classes, with 50,000 training images and 10,000 test images.
- **CIFAR-100** (Krizhevsky et al., 2009): 60,000 natural images from 100 classes, split into 50,000 training and 10,000 test images.
- **DTD-47** (Cimpoi et al., 2014): 5,640 texture images from 47 classes, used for texture classification tasks.
- **Caltech-101** (Fei-Fei et al., 2004): 9,146 images from 101 object classes plus a background class, with varying numbers of images per class.
- **CUB-200** (Wah et al., 2011): 11,788 bird images from 200 species, with standard splits for training and testing.
- **Oxford Flowers-102** (Nilsback & Zisserman, 2008): 8,189 images from 102 flower classes, with 6,149 training images, 1,020 validation images, and 1,020 test images.
- **Stanford Cars-196** (Krause et al., 2013): 16,185 car images from 196 models, split into 8,144 training images and 8,041 test images.
- **ImageNet-1K** (Deng et al., 2009): 1.28 million training images and 50,000 validation images across 1,000 classes, following standard splits for large-scale image classification.
- **AG News-4** (Zhang et al., 2015): a text classification dataset containing 120,000 training and 7,600 test samples across 4 news classes.

### B.2 IMPLEMENTATION DETAILS

We implement the proposed NAF using the TLlib library (Jiang et al., 2022) and apply weak and strong augmentation techniques (Cubuk et al., 2020) in the target domain. All experiments are conducted on NVIDIA V100 series GPUs. For image classification, we implement the representation extractor $g_t$ using ResNet (He et al., 2016) backbones pre-trained on ImageNet-1K for all datasets (except for ImageNet-1K itself, where the backbone is trained from scratch). As for text classification, we employ the pre-trained BERT model (Devlin et al., 2019) as the text encoder. The noise projector $g_n$ is a non-linear layer with ReLU activation (Nair & Hinton, 2010), and the classifier $f$ is a single linear layer. Furthermore, we utilize mini-batch SGD with a momentum of 0.9 as the optimizer, setting batch sizes to 32 for CIFAR-10, CIFAR-100, DTD-47, Caltech-101, CUB-200, Oxford Flowers-102, and Standard Cars-196, and 128 for ImageNet-1K.

In NAF, it is necessary to calculate the class mean for each class. To address the mini-batch issue, we follow (Xie et al., 2018) and employ an *exponential moving average* to update the class means as follows: $\mathbf{m}_n^c = (1 - \lambda) \cdot \mathbf{m}_o^c + \lambda \cdot \mathbf{m}_b^c$, where $\mathbf{m}_o^c$ and $\mathbf{m}_n^c$ denote the previous and updated $c$-th class means, respectively, and $\mathbf{m}_b^c$ is the $c$-th class mean calculated from the current mini-batch. Table 8 summarizes the detailed parameter configurations used in this paper.

Table 8: Detailed parameter configuration used in this paper.

| Method | Dataset | Backbone | $\alpha$ | $\beta$ | $\lambda$ | learning rate | iterations |
|---|---|---|---|---|---|---|---|
| NAF | CIFAR-10 / DTD-47 | ResNet-50 / ResNet-18 | 1 | 1 | | 0.03 | |
| | CIFAR-100 | ResNet-50 / ResNet-18 | 10 | 10 | | 0.01 | 10,000 |
| | Caltech-101 | ResNet-50 / ResNet-18 | 1 | 10 | | 0.003 | |
| | CUB-200 | ResNet-18 | 1 | 50 | 0.7 | 0.003 | 8,000 |
| | Oxford Flowers-102 / Stanford Cars-196 | ResNet-18 | 1 | 50 | | 0.03 | 4,000 / 6,000 |
| | ImageNet-1K | ResNet-18 | 0.1 | 10 | | 0.01 | 80,000 |
| LERM + NAF | CIFAR-10 | | 1 | 1 | 0.99 | 0.03 | |
| | CIFAR-10 / CIFAR-100 | ResNet-18 | 10 | 10 | 0.99 / 0.7 | 0.03 / 0.01 | 10,000 |
| Others + NA | DTD-47 | | 1 | 5 | 0.7 | 0.03 | |
| | Caltech-101 | | 1 | 10 | 0.7 | 0.003 | |

## C  NOTATION AND PROOF OF THEOREM 1

### C.1  NOTATION

For clarity, Table 9 summarizes the notations used in this paper.

Table 9: A summary of the notations used in this paper.

| Notation | Description |
|---|---|
| $C$ | Total number of classes |
| $\mathcal{C}$ | Class index set $\{0, \ldots, C-1\}$ |
| $\mathcal{D}_l, \mathcal{D}_u$ | Labeled and unlabeled target sample sets |
| $\mathcal{D}_e$ | Test target sample set (used only for evaluation) |
| $\mathcal{D}_t$ | Target domain: $\mathcal{D}_l \cup \mathcal{D}_u \cup \mathcal{D}_e$ |
| $\mathcal{D}_n$ | Noise domain |
| $\mathbf{x}_i^l, \mathbf{x}_i^u, \mathbf{x}_i^e$ | $i$-th sample from $\mathcal{D}_l, \mathcal{D}_u$, and $\mathcal{D}_e$, respectively |
| $y_i^l$ | Label of $\mathbf{x}_i^l, y_i^l \in \mathcal{C}$ |
| $\mathbf{n}_i$ | $i$-th noise in $\mathcal{D}_n$ |
| $y_i$ | Label of $\mathbf{n}_i, y_i \in \mathcal{C}$ |
| $\mathcal{X}$ | Sample space (e.g., a pixel-level image space) |
| $\mathcal{E}$ | Noise space (e.g., a $p$-dimensional space) |
| $\mathcal{Z}$ | Domain-shared representation space |
| $\mathcal{F}$ | Hypothesis space over $\mathcal{Z}$ |
| $\widetilde{\mathcal{P}}_t$ | Target distribution over $\mathcal{Z}$ |
| $\widetilde{\mathcal{P}}_n$ | Noise distribution over $\mathcal{Z}$ |
| $\mathcal{U}_n, \mathcal{U}_t$ | Unlabeled sample sets drawn from $\widetilde{\mathcal{P}}_n$ and $\widetilde{\mathcal{P}}_t$, respectively |
| $\mathcal{L}_n, \mathcal{L}_t$ | Labeled sample sets drawn from $\widetilde{\mathcal{P}}_n$ and $\widetilde{\mathcal{P}}_t$, respectively |
| $g_t(\cdot)$ | Representation extractor for target samples |
| $g_n(\cdot)$ | Noise projector for noise |
| $f(\cdot)$ | Domain-shared classifier |
| $n_l, n_u$ | Number of labeled and unlabeled target samples |
| $n$ | Number of noise |

### C.2  PROOF OF THEOREM 1

**Theorem 1.** *(Generalization Bound of SSNA) Let $\hat{f} = \arg\min_{f \in \mathcal{F}} \hat{\epsilon}_\alpha(f)$ be the empirical minimizer of $\hat{\epsilon}_\alpha(f)$, and let $f_t^* = \arg\min_{f \in \mathcal{F}} \epsilon_t(f)$ be the target error minimizer. Then, for any $\delta \in (0,1)$, with probability at least $1 - \delta$ (over the choice of the samples), we have:*

$$\epsilon_t(\hat{f}) \leq \epsilon_t(f_t^*) + \mathcal{O}\left(\gamma\sqrt{\frac{d\log m + \log(\frac{1}{\delta})}{m}}\right) + 2(1-\alpha)\left[\frac{1}{2}\hat{d}_{\mathcal{H}\Delta\mathcal{H}}(\mathcal{U}_n,\mathcal{U}_t) + \mathcal{O}\left(\sqrt{\frac{d\log m' + \log(\frac{1}{\delta})}{m'}}\right)\right.$$

$$\left. + \hat{\epsilon}_n(\hat{f}) + \hat{\epsilon}_t(\hat{f}) + \mathcal{O}\left(\sqrt{\frac{d\log(\frac{(1-\beta)m}{d}) + \log(\frac{1}{\delta})}{(1-\beta)m}}\right) + \mathcal{O}\left(\sqrt{\frac{d\log(\frac{\beta m}{d}) + \log(\frac{1}{\delta})}{\beta m}}\right)\right],$$

*where $\gamma = \sqrt{\frac{\alpha^2}{\beta} + \frac{(1-\alpha)^2}{1-\beta}}$, and $\hat{d}_{\mathcal{H}\Delta\mathcal{H}}(\mathcal{U}_n, \mathcal{U}_t)$ is the empirical $\mathcal{H}$-divergence estimated from noise and target samples in $\mathcal{Z}$.*

We now outline the main steps of the proof, based on (Ben-David et al., 2010), beginning with Lemmas 1 and 2, which correspond to Lemmas 4 and 5 in (Ben-David et al., 2010).

**Lemma 1.** *Let $f$ be a hypothesis in hypothesis space $\mathcal{F}$. Then $|\epsilon_\alpha(f) - \epsilon_t(f)| \leq (1-\alpha)\left(\frac{1}{2}d_{\mathcal{H}\Delta\mathcal{H}}(\widetilde{\mathcal{P}}_n, \widetilde{\mathcal{P}}_t) + \lambda\right)$, where $\lambda := \min_{f \in \mathcal{F}} \epsilon_n(f) + \epsilon_t(f)$.*

**Lemma 2.** *For a fixed hypothesis $f$, if $m$ random labeled samples are drawn, with $\beta m$ from $\widetilde{\mathcal{P}}_t$ and $(1-\beta)m$ from $\widetilde{\mathcal{P}}_n$, then for any $\delta \in (0,1)$, with probability at least $1 - \delta$ (over the choice of*

*samples), we have:*

$$\Pr\left[|\hat{\epsilon}_\alpha(f) - \epsilon_\alpha(f)| \geq \epsilon\right] \leq 2\exp\left(\frac{-2m\epsilon^2}{\frac{\alpha^2}{\beta} + \frac{(1-\alpha)^2}{1-\beta}}\right). \tag{7}$$

For brevity, we omit the proofs of Lemmas 1 and 2 here, which are available in (Ben-David et al., 2010). Next, we provide a detailed proof for Theorem 1.

*Proof.* In the proof below, steps labeled L1 and L2 correspond to applications of Lemma 1 and Lemma 2, respectively, with L2 additionally employing standard techniques of sample symmetrization and VC-dimension–based growth-function bounds (Anthony & Bartlett, 1999).

$$\epsilon_t(\hat{f}) \leq \epsilon_\alpha(\hat{f}) + (1-\alpha)\left(\frac{1}{2}d_{\mathcal{H}\Delta\mathcal{H}}(\widetilde{\mathcal{P}}_n, \widetilde{\mathcal{P}}_t) + \lambda\right) \text{(L1)} \tag{8}$$

$$\leq \hat{\epsilon}_\alpha(\hat{f}) + 2\gamma\sqrt{\frac{2d\log(2(m+1)) + 2\log(\frac{16}{\delta})}{m}} + (1-\alpha)\left(\frac{1}{2}d_{\mathcal{H}\Delta\mathcal{H}}(\widetilde{\mathcal{P}}_n, \widetilde{\mathcal{P}}_t) + \lambda\right) \text{(L2)} \tag{9}$$

$$\leq \hat{\epsilon}_\alpha(f_t^*) + 2\gamma\sqrt{\frac{2d\log(2(m+1)) + 2\log(\frac{16}{\delta})}{m}} + (1-\alpha)\left(\frac{1}{2}d_{\mathcal{H}\Delta\mathcal{H}}(\widetilde{\mathcal{P}}_n, \widetilde{\mathcal{P}}_t) + \lambda\right) \tag{10}$$

$$\leq \epsilon_\alpha(f_t^*) + 4\gamma\sqrt{\frac{2d\log(2(m+1)) + 2\log(\frac{16}{\delta})}{m}} + (1-\alpha)\left(\frac{1}{2}d_{\mathcal{H}\Delta\mathcal{H}}(\widetilde{\mathcal{P}}_n, \widetilde{\mathcal{P}}_t) + \lambda\right) \text{(L2)} \tag{11}$$

$$\leq \epsilon_t(f_t^*) + 4\gamma\sqrt{\frac{2d\log(2(m+1)) + 2\log(\frac{16}{\delta})}{m}} + 2(1-\alpha)\left(\frac{1}{2}d_{\mathcal{H}\Delta\mathcal{H}}(\widetilde{\mathcal{P}}_n, \widetilde{\mathcal{P}}_t) + \lambda\right) \text{(L1)} \tag{12}$$

$$\leq \epsilon_t(f_t^*) + 4\gamma\sqrt{\frac{2d\log(2(m+1)) + 2\log(\frac{16}{\delta})}{m}}$$
$$+ 2(1-\alpha)\left(\frac{1}{2}\hat{d}_{\mathcal{H}\Delta\mathcal{H}}(\mathcal{U}_n, \mathcal{U}_t) + 4\sqrt{\frac{2d\log(2m') + \log\left(\frac{8}{\delta}\right)}{m'}} + \lambda\right) \tag{13}$$

$$\leq \epsilon_t(f_t^*) + 4\gamma\sqrt{\frac{2d\log(2(m+1)) + 2\log(\frac{16}{\delta})}{m}}$$
$$+ 2(1-\alpha)\left(\frac{1}{2}\hat{d}_{\mathcal{H}\Delta\mathcal{H}}(\mathcal{U}_n, \mathcal{U}_t) + 4\sqrt{\frac{2d\log(2m') + \log\left(\frac{8}{\delta}\right)}{m'}} + \epsilon_n(\hat{f}) + \epsilon_t(\hat{f})\right) \tag{14}$$

$$\leq \epsilon_t(f_t^*) + 4\gamma\sqrt{\frac{2d\log(2(m+1)) + 2\log(\frac{16}{\delta})}{m}}$$
$$+ 2(1-\alpha)\left(\frac{1}{2}\hat{d}_{\mathcal{H}\Delta\mathcal{H}}(\mathcal{U}_n, \mathcal{U}_t) + 4\sqrt{\frac{2d\log(2m') + \log\left(\frac{8}{\delta}\right)}{m'}}\right.$$
$$\left. + \hat{\epsilon}_n(\hat{f}) + \hat{\epsilon}_t(\hat{f}) + \sqrt{\frac{8d\log(\frac{2e(1-\beta)m}{d}) + 8\log(\frac{16}{\delta})}{(1-\beta)m}} + \sqrt{\frac{8d\log(\frac{2e\beta m}{d}) + 8\log(\frac{16}{\delta})}{\beta m}}\right). \tag{15}$$

Accordingly, we have:

$$\epsilon_t(\hat{f}) \leq \epsilon_t(f_t^*) + \mathcal{O}\left(\gamma\sqrt{\frac{d\log m + \log(\frac{1}{\delta})}{m}}\right) + 2(1-\alpha)\left[\frac{1}{2}\hat{d}_{\mathcal{H}\Delta\mathcal{H}}(\mathcal{U}_n, \mathcal{U}_t) + \mathcal{O}\left(\sqrt{\frac{d\log m' + \log(\frac{1}{\delta})}{m'}}\right)\right.$$
$$\left. + \hat{\epsilon}_n(\hat{f}) + \hat{\epsilon}_t(\hat{f}) + \mathcal{O}\left(\sqrt{\frac{d\log(\frac{(1-\beta)m}{d}) + \log(\frac{1}{\delta})}{(1-\beta)m}}\right) + \mathcal{O}\left(\sqrt{\frac{d\log(\frac{\beta m}{d}) + \log(\frac{1}{\delta})}{\beta m}}\right)\right]. \tag{16}$$

Eq. (10) holds due to $\hat{f} = \arg\min_{f \in \mathcal{F}} \hat{\epsilon}_\alpha(f)$, Eq. (13) is established using the bound proposed in (Ben-David et al., 2010), Eq. (14) holds because $\lambda := \min_{f \in \mathcal{F}} \epsilon_n(f) + \epsilon_t(f) \le \epsilon_n(\hat{f}) + \epsilon_t(\hat{f})$, and Eq. (15) uses the bound from (Mohri et al., 2018).

## D    SUPPLEMENTARY EXPERIMENTAL RESULTS

We provide additional results for SOTA + NAF on DTD-47 and Caltech-101 using ResNet-18. As shown in Table 10, SOTA + NAF consistently outperforms the standalone SOTA methods across most scenarios, further demonstrating the effectiveness of NAF in leveraging the noise domain to enhance the performance of the target domain.

Table 10: Accuracy (%) comparison on DTD-47 and Caltech-101 using ResNet-18. Here, $\Delta$ indicates the performance gain introduced by NAF.

| Datasets | DTD-47 | | | | | Caltech-101 | | | | |
|---|---|---|---|---|---|---|---|---|---|---|
| Epoch | 5 | 10 | 15 | 20 | Average | 5 | 10 | 15 | 20 | Average |
| UDA (Xie et al., 2020) | 46.28 | 46.81 | 46.90 | 47.32 | 46.83 | 79.20 | 79.61 | 80.00 | 80.28 | 79.77 |
| UDA + NAF | 46.88 | 47.89 | 49.10 | 49.22 | 48.27 | 80.98 | 81.40 | 81.21 | 81.43 | 81.26 |
| $\Delta$ | +0.60 | +1.08 | +2.20 | +1.90 | +1.44 | +1.78 | +1.79 | +1.21 | +1.15 | +1.49 |
| FixMatch (Sohn et al., 2020) | 46.51 | 47.78 | 48.09 | 48.23 | 47.65 | 80.13 | 80.27 | 80.28 | 79.99 | 80.17 |
| FixMatch + NAF | 48.85 | 49.57 | 50.12 | 49.86 | 49.60 | 80.96 | 80.96 | 80.42 | 80.42 | 80.69 |
| $\Delta$ | +2.34 | +1.79 | +2.03 | +1.63 | +1.95 | +0.83 | +0.69 | +0.14 | +0.43 | +0.52 |
| FlexMatch (Zhang et al., 2021) | 50.66 | 51.29 | 50.94 | 50.69 | 50.90 | 82.74 | 83.83 | 83.61 | 83.70 | 83.47 |
| FlexMatch + NAF | 50.51 | 50.87 | 51.03 | 51.35 | 50.94 | 83.22 | 84.08 | 83.74 | 83.77 | 83.70 |
| $\Delta$ | -0.15 | -0.42 | +0.09 | +0.66 | +0.04 | +0.48 | +0.25 | +0.13 | +0.07 | +0.23 |
| DebiasMatch (Wang et al., 2022) | 45.67 | 45.99 | 45.46 | 46.42 | 45.89 | 80.87 | 81.09 | 81.29 | 81.60 | 81.21 |
| DebiasMatch + NAF | 49.01 | 49.79 | 50.02 | 50.09 | 49.73 | 82.46 | 82.62 | 82.77 | 82.60 | 82.61 |
| $\Delta$ | +3.34 | +3.80 | +4.56 | +3.67 | +3.84 | +1.59 | +1.53 | +1.48 | +1.00 | +1.40 |
| DST (Chen et al., 2022) | 49.84 | 51.68 | 52.27 | 51.93 | 51.43 | 80.75 | 81.85 | 82.19 | 82.16 | 81.74 |
| DST + NAF | 51.08 | 52.00 | 52.54 | 52.55 | 52.04 | 81.70 | 82.72 | 82.85 | 82.87 | 82.54 |
| $\Delta$ | +1.24 | +0.32 | +0.27 | +0.62 | +0.61 | +0.95 | +0.87 | +0.66 | +0.71 | +0.80 |
| LERM (Zhang et al., 2024) | 47.20 | 47.50 | 48.03 | 48.42 | 47.79 | 82.36 | 83.06 | 82.98 | 83.13 | 82.88 |
| LERM + NAF | 48.85 | 48.83 | 48.87 | 48.92 | 48.87 | 83.14 | 83.59 | 83.23 | 83.06 | 83.26 |
| $\Delta$ | +1.65 | +1.33 | +0.84 | +0.50 | +1.08 | +0.78 | +0.53 | +0.25 | -0.07 | +0.38 |

## E    ADDITIONAL ANALYSIS EXPERIMENTS

**Q13. How does the performance of using noise as a source domain compare to that of using real samples?** We investigate this question on the Office-Caltech-10 dataset, which is a transfer learning benchmark containing 10 shared object classes from Office-31 (Saenko et al., 2010) and Caltech-256 (Griffin et al., 2007). Caltech is used as the target domain, where 4 labeled samples per class are randomly selected and the rest are treated as unlabeled. For the source domain, we consider two settings: a synthetic noise domain (denoted as NAF (Noise)) and the Amazon domain (denoted as NAF (Real)). For each source domain, we vary the number of labeled samples per class among 10, 20, 30, 40, and 50. Table 11 reports the results, from which we make the following observations. (1) Both NAF (Noise) and NAF (Real) outperform ERM, and NAF (Real) performs slightly better, indicating that even synthetic noise can effectively guide the learning of the target samples without access to real samples. (2) Even a limited number of source samples significantly improves performance, as they can form a class-discriminative structure that achieves positive transfer regardless of whether the samples are real or synthetic. Those findings together support the conclusion that synthetic noise can serve as a practical substitute when real out-of-domain samples are unavailable.

**Q14. How does the amount of noise impact NAF?** We vary the amount of noise per class (*i.e.*, 0, 10, 50, 100, 200) to evaluate its impact on NAF. The results on CIFAR-100 using ResNet-18 are shown in Figure 5a. As can be observed, when the amount of noise is zero, NAF degenerates to ERM, resulting in poor performance. As the noise increases from 10 to 100, performance remains relatively

Table 11: Accuracy (%) comparison on Amazon-to-Caltech-10 transfer task using ResNet-18 with different number of source samples.

| # source samples per class | 10 | 20 | 30 | 40 | 50 |
|---|---|---|---|---|---|
| ERM | 83.51 | 83.51 | 83.51 | 83.51 | 83.51 |
| NAF (Noise) | 89.89 | 88.65 | 88.83 | 88.12 | 89.36 |
| NAF (Real) | 90.25 | 90.07 | 90.96 | 92.20 | 91.14 |

stable, indicating that the presence of a class-discriminative structure in the noise domain is more important than the total amount of noise. Even a small number of noise samples can form separable patterns in the shared representation space and guide the alignment of target representations. When the noise per class reaches 200, performance slightly declines, suggesting that excessive noise may increase learning difficulty and provide limited additional benefit.

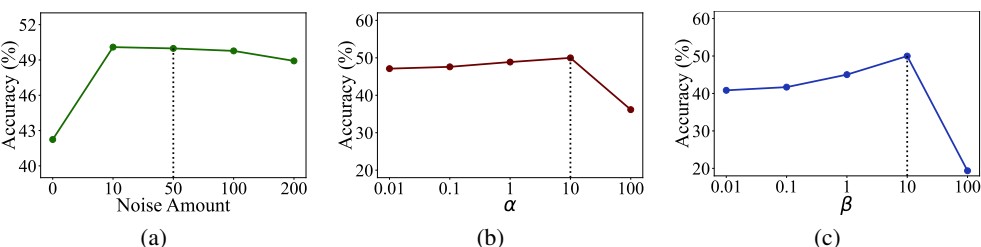

(a)  (b)  (c)

Figure 5: Accuracy (%) comparison on CIFAR-100 using ResNet-18 with varying (a) amounts of noise, (b) values of $\alpha$, and (c) values of $\beta$.

**Q15. How do the hyperparameters $\alpha$ and $\beta$ influence NAF?** We analyze the sensitivity of $\alpha$ and $\beta$ on CIFAR-100 using ResNet-18. Figures 5b and 5c present the performance of NAF under varying values of $\alpha$ and $\beta$, respectively. The results show that NAF performs well and remains relatively stable when $\alpha$ and $\beta$ are close to the default value of 10. However, when either parameter increases to 100, a significant performance drop is observed, suggesting that excessive focus on the noise domain hurts the performance of the target domain.

**Q16. What is the impact of using class means for noise construction on model performance?** Using class means as the noise domain represents a special case of noise construction, where all noise within a class collapses to a single class mean. To investigate its effect, we consider two variants: NAF with Fixed Class Means, *i.e.*, NAF (FCM), and NAF with Learned Class Means, *i.e.*, NAF (LCM). In NAF (FCM), class means in the noise domain are initialized as orthogonal vectors and remain fixed during training. In NAF (LCM), class means are similarly initialized but updated during training through the noise projector. Table 12 reports the results on CIFAR-100 with 4 labeled samples using ResNet-18. We have several insightful observations. (1) Both NAF (FCM) and NAF (LCM) outperform ERM, indicating that positive transfer can still occur even when the noise domain is simplified to class means. The reason is that class means retain the separability among categories, thereby preserving a discriminative structure that provides useful guidance for aligning target representations. (2) NAF (LCM) achieves an accuracy of 47.72%, outperforming NAF (FCM) (46.68%) by 1.04%, demonstrating that using learnable noise may be more effective than using fixed noise. (3) NAF achieves 49.98% accuracy, surpassing NAF (LCM), highlighting that different noise construction strategies lead to varying levels of discriminative structure, which in turn critically influences alignment and overall performance.

Table 12: Accuracy (%) comparison of different noise construction strategies on CIFAR-100 using ResNet-18.

| ERM | NAF (FCM) | NAF (LCM) | NAF |
|---|---|---|---|
| 42.24 | 46.68 | 47.72 | 49.98 |

**Q17. How does NAF perform under varying inter-class distances in the noise domain?** We perform ablation studies by constructing noise domains with controlled inter-class distances. Specifically, we first sample a global mean $\boldsymbol{\mu}$ and class-specific offsets $\boldsymbol{\epsilon}_c$ from a standard Gaussian distribution, and define class means as $\boldsymbol{\mu}_c = \boldsymbol{\mu} + \delta\boldsymbol{\epsilon}_c$, where $\delta$ explicitly controls the distance between class means. Then, we sample 50 noise per class from the Gaussian distribution $\mathcal{N}(\boldsymbol{\mu}_c, \mathbf{I})$. By varying $\delta$ over the set $\{0, 0.1, 0.3, 0.5, 1\}$, we adjust the inter-class distances of the noise domain, corresponding to Jensen–Shannon (JS) divergence values of 0, 2.57, 23.16, 64.33, and 257.33, respectively. Higher JS divergence values indicate larger inter-class distances. The results on CIFAR-100 using ResNet-18 are reported in Table 13. We can see that when $\delta = 0$, NAF performs comparably to ERM. As the value of $\delta$ increases, the performance improves accordingly, indicating that larger inter-class distances in the noise domain lead to enhanced generalization performance.

Table 13: Accuracy (%) comparison on CIFAR-100 using ResNet-18 with varying inter-class distances of the noise domain.

| $\delta$ | 0 | 0.1 | 0.3 | 0.5 | 1 |
|---|---|---|---|---|---|
| ERM | 42.24 | 42.24 | 42.24 | 42.24 | 42.24 |
| NAF | 43.80 | 43.78 | 46.57 | 49.57 | 49.78 |

**Q18. How does NAF compare with plug-in modules for SSL?** LERM (Zhang et al., 2024) is an effective plug-in module for SSL. We compare NAF and LERM under both ERM and DST on CIFAR-10 using ResNet-18. As shown in Table 14, NAF provides a larger improvement than LERM when combined with ERM. On DST, both methods yield modest improvements, with NAF showing comparable performance to LERM. Those results suggest that NAF may serve as a competitive plug-in module for SSL.

Table 14: Accuracy (%) comparison of ERM and DST combined with either LERM or NAF on CIFAR-10 using ResNet-18.

| Method | Base | +LERM | +NAF |
|---|---|---|---|
| ERM | 58.15 | 64.90 | 71.83 |
| DST | 84.96 | 86.82 | 86.58 |

**Q19. How does NAF compare with contrastive learning methods?** We compare NAF with a contrastive learning method inspired by CLIP (Radford et al., 2021), referred to as CL. CL applies a contrastive loss between weakly-augmented and strongly-augmented unlabeled target samples to encourage consistent representations for different augmentations of the same sample. On CIFAR-100 using ResNet-18, ERM + CL achieves an accuracy of 44.15%, improving over ERM alone (42.24%) but remaining lower than ERM + NAF (49.98%). One potential reason is that CL only uses the unlabeled target samples for contrastive learning but does not leverage their pseudo-labels.

## F  DECLARATION OF USE OF LARGE LANGUAGE MODELS

In this paper, large language models are used solely to assist with writing, improving clarity, phrasing, and presentation.

