# OpenReview forum: "Semi-Supervised Noise Adaptation: Transferring Knowledge from Noise Domain"
_ICLR.cc/2026/Conference — Submitted to ICLR 2026_

### Official Review · Reviewer_GdmS · 2025-10-26

**Soundness:** 4
**Presentation:** 4
**Contribution:** 3
**Rating:** 6
**Confidence:** 3

**Summary:**

This paper introduces a novel solution for Semi-Supervised Noise Adaptation (SSNA), and proposes a new Noise Adaptation Framework (NAF) that leverages a synthetic noise domain, that is, constructed from random distributions to improve generalisation in a semi-supervised target domain. The authors proof a theoretical generalisation bound inspired by domain adaptation theory and design a learning objective combining supervised, noise-domain, and alignment losses. Experiments on CIFAR-10/100, DTD, Caltech-101, and ImageNet-1K show consistent improvements over Empirical Risk Minimisation and compatibility with existing semi-supervised learning methods.

**Strengths:**

Interesting improvement of existing concept: The idea of transferring structure from a synthetic, non-semantic noise domain is novel and could inspire future research on structure-based generalisation.

Clear theoretical definition and framing: The authors connect the proposed method to domain adaptation bounds, offering a explanation for why structured noise might help generalisation.

Extensive experiments: The empirical section is broad and thorough, showing improvements across multiple benchmarks.
Good writing and reproducibility: The paper is well written, clearly organised, and accompanied by public code, which is appreciated.

**Weaknesses:**

The proposed method is a extension of prior work "Noise May Contain Transferable Knowledge: Understanding Semi-supervised Heterogeneous Domain Adapta…", with some new experiments and explanation, however the visualisation used for explanation is almost identical (i.e. Figure 1 of this paper vs Figure 4 of the referred paper)

SSNA made a strong assumption on a one-to-one mapping between noise classes and target classes and a known number of target classes. This can be limited in the setting like imbalanced dataset. And the paper didn't not discuss the constraints or experiment with dataset as such.

In most of the experiment results, only the "accuracy" metrics are presented, this can be misleading if the dataset is imbalanced, and does not provide error type. Potentially need to provide other metrics like recall or f1.

Only random noise were discussed and experimented, would other type of noise like Gaussian, uniform, structured noise also applicable?

The paper does not compare to modern self-supervised or contrastive regularisation methods that might yield similar improvements without requiring a noise domain.

**Questions:**

Overall, I'm still slightly confused on the method is trying to minimised the error on the randomly labelled noise, but there is no clear link to show reduce the error on noise lead to lower error on the real target domain? This can be regularisation by add "noise" to the training instead of really "transferring" learning as it claim?

---

> ### Author Response · Authors · 2025-11-25
> **Part I**
>
> We sincerely appreciate your thoughtful review and valuable feedback. We address your comments point by point.
>
> >1. The proposed method is a extension of prior work "Noise May Contain Transferable Knowledge: Understanding Semi-supervised Heterogeneous Domain Adapta…", with some new experiments and explanation, however the visualisation used for explanation is almost identical (i.e. Figure 1 of this paper vs Figure 4 of the referred paper)
>
> **A1**: We appreciate your careful comparison. To better illustrate the SSNA setting, we have redesigned Figure 1 with a new layout and style. Please see Figure 1 in the revision.
>
> >2. SSNA made a strong assumption on a one-to-one mapping between noise classes and target classes and a known number of target classes. This can be limited in the setting like imbalanced dataset. And the paper didn't not discuss the constraints or experiment with dataset as such. In most of the experiment results, only the "accuracy" metrics are presented, this can be misleading if the dataset is imbalanced, and does not provide error type. Potentially need to provide other metrics like recall or f1.
>
>
> **A2**: Thank you for your suggestion. To investigate the performance of NAF under class imbalance in the target domain, we conduct an experiment on CIFAR-10 using a long-tailed setup. In this configuration, the labeled and unlabeled sets have per-class sample counts of [50, 30, 20, 10, 6, 4, 3, 2, 2, 1] and [1000, 600, 200, 100, 60, 40, 20, 10, 6, 4], respectively. NAF achieves an accuracy of 56.38\% and a macro F1-score of 53.22\%, outperforming ERM, which attains 51.19\% accuracy and a macro F1-score of 45.73\%. Those results suggest that NAF remains effective even under such a class imbalance scenario. We have included this experiment into Q11 in the revision.
>
> >3. Only random noise were discussed and experimented, would other type of noise like Gaussian, uniform, structured noise also applicable?
>
> **A3**: We appreciate your suggestion. To evaluate the effectiveness of NAF under different noise generation strategies, we conduct additional experiments by varying the noise along three dimensions. (1) **Covariance Scale**: In the original setup, we first sample a mean for each class from a standard Gaussian distribution. Next, for each class, we generate individual noise from a Gaussian distribution with the corresponding mean and identity covariance $\mathbf{I}$. We additionally evaluate two configurations in which all class covariances are scaled to $0.1 \cdot \mathbf{I}$ and $10 \cdot \mathbf{I}$. (2) **Noise Dimensionality**: In the original setup, the noise dimensionality is set to 1024. We additionally evaluate two configurations with noise dimensionalities of 512 and 2048. (3) **Distribution Type**: In the original setup, the noise is drawn from a Gaussian distribution. We additionally test the Log-normal distribution and the Laplace distribution. The results, listed in the following table, indicate that NAF achieves comparable performance across a variety of noise settings, including variations in covariance scale, noise dimensionality, and distribution type. Those observations suggest that NAF can accommodate different noise configurations, highlighting its potential flexibility. We have included this experiment into Q10 in the revision.
>
> Table 1. Accuracy (%) comparison on CIFAR-100 using ResNet-18 with noise drawn from various noise generation strategies. Here, $\boldsymbol{\mu}_c$ is the class mean belonging to class $c$, and $d$ is the dimensionality of the noise.
> | Noise Configuration | Noise Distribution | Accuracy (%) |
> |---------------------------|-------------------------------------|--------------|
> | **Baseline**      | Gaussian: $\mathcal{N}(\boldsymbol{\mu}_c, \mathbf{I}), d = 1024$ | 49.98|
> | **Covariance Scale**    | Gaussian: $\mathcal{N}(\boldsymbol{\mu}_c, 0.1 \cdot \mathbf{I}), d = 1024$ | 50.38 |
> |                   | Gaussian: $\mathcal{N}(\boldsymbol{\mu}_c, 10 \cdot \mathbf{I}), d = 1024$ | 47.64  |
> | **Noise Dimensionality**| Gaussian: $\mathcal{N}(\boldsymbol{\mu}_c, \mathbf{I}), d = 512$ | 49.44   |
> |                   | Gaussian: $\mathcal{N}(\boldsymbol{\mu}_c, \mathbf{I}), d = 2048$ | 51.04  |
> | **Distribution Type**| Log-normal: $\log \mathcal{N}(\boldsymbol{\mu}_c, \mathbf{I}), d = 1024$ | 48.31 |
> |                      | Laplace: $\mathcal{L}((\boldsymbol{\mu}_c)_d, 1/\sqrt{2}), d = 1024$ | 49.99 |

---

> ### Author Response · Authors · 2025-11-25
> **Part II**
>
> > 4. The paper does not compare to modern self-supervised or contrastive regularisation methods that might yield similar improvements without requiring a noise domain.
>
> **A4**: Thank you for the suggestion. We conduct an additional experiment using a contrastive learning method inspired by CLIP [1], referred to as CL. CL applies a contrastive loss between weakly-augmented and strongly-augmented unlabeled target samples to encourage consistent representations for different augmentations of the same sample. On CIFAR-100 using ResNet-18, ERM + CL achieves an accuracy of 44.15%, improving over ERM alone (42.24%) but remaining lower than ERM + NAF (49.98%). One potential reason is that CL only uses the unlabeled target samples for contrastive learning but does not leverage their pseudo-labels. We have included this experiment into Q19 in the revision.
>
> [1] Radford A et al. Learning transferable visual models from natural language supervision. ICML, 2021.
>
> >Q1. I'm still slightly confused on the method is trying to minimised the error on the randomly labelled noise, but there is no clear link to show reduce the error on noise lead to lower error on the real target domain? This can be regularisation by add "noise" to the training instead of really "transferring" learning as it claim?
>
> **AQ1**: Thank you for the question. As shown in Theorem 1, the expected target error is upper-bounded by several terms, including the empirical errors on the two domains and their empirical distributional discrepancy. Consequently, minimizing the noise error alone does not directly reduce the target error; jointly reducing the relevant terms leads to a tighter upper bound. Furthermore, NAF leverages the class-discriminative structure in the noise domain by aligning it with the target representations to guide learning on the target domain. Empirically, as evaluated in Q8 of the submission (Q9 in the revision), when we remove the class structure in the noise domain (e.g., by using a single point for all classes), the performance drops below that of ERM. This observation suggests that NAF leverages the class-discriminative structure in the noise domain to facilitate better generalization in the target domain. We have included this clarification in the revision; please refer to Lines 237-241 and 467–468.

---

> > ### Comment · Reviewer_GdmS · 2025-11-27
> >
> > The rebuttal addresses my concerns; I will keep my current rating and confidence.

---

> > > ### Author Response · Authors · 2025-11-27
> > >
> > > We sincerely thank you for your thoughtful and positive assessment of our work.

---

### Official Review · Reviewer_Gj7X · 2025-10-27

**Soundness:** 4
**Presentation:** 4
**Contribution:** 3
**Rating:** 8
**Confidence:** 4

**Summary:**

Unlike conventional transfer learning where the source domain consists of semantically meaningful data, SSNA leverages a noise domain generated from Gaussian distributions as a surrogate source domain to assist target-domain learning. The authors theoretically derive a generalization error bound for the target domain and propose the NAF, which tightens this bound by jointly minimizing the empirical risk on the target domain, the noise-domain risk, and the inter-domain distributional discrepancy. Experiments conducted on multiple benchmark datasets demonstrate that NAF achieves significant performance improvements compared with various existing methods.

**Strengths:**

- The paper introduces the SSNA problem, which treats the noise domain as a surrogate source domain. This setting holds potential application value in scenarios with privacy constraints or data scarcity.
- In Section 4.1, the authors derive a generalization bound (Theorem 1), explicitly showing that the bound depends on three components: the empirical error on the target domain, the empirical error on the noise domain, and the H-divergence between them.
- The code and parameter configurations are detailed in Appendix B, and an open-source link is provided to enhance reproducibility.

**Weaknesses:**

- There are concerns regarding the assumptions and some conclusions of NAF. [1] demonstrates that deep neural networks trained on completely random labels or random inputs can fully memorize arbitrary input–output mappings. [2] similarly finds that during optimization, deep networks with structural priors such as convolutional weight sharing, batch normalization, and nonlinear mappings tend to enforce linearly separable clustering in feature space.
Considering these findings, the reviewer is concerned that the “discriminative structure formed in the noise domain” claimed by NAF may in fact result from pseudo-structural effects induced by the inductive bias of deep models like ResNet, rather than from the noise distribution itself.

- There are also concerns about the way the noise domain is generated. The paper uses a Gaussian distribution, but in real-world settings, natural noise (e.g., Poisson or long-tailed distributions) often exhibits non-Gaussian characteristics. Under such conditions, can the proposed noise projector (g_n) and distribution alignment mechanism still effectively extract discriminative structures? Since non-Gaussian noise tends to have larger intra-class variance, forming compact clusters within each class may be difficult, potentially weakening the effectiveness of knowledge transfer.

- Would the noise distribution parameters (e.g., variance σ, dimensionality d) affect generalization performance?

[1] Understanding deep learning requires rethinking generalization, ICLR'17

[2] Learning to See by Looking at Noise, NIPS'21

**Questions:**

This paper addresses a highly interesting and novel problem. The writing and presentation are excellent, and the figures and results are consistent with the claims.

However, some issues may influence the final rating:

- Are there any visualizations, experiments, or theoretical analyses demonstrating that the transferable discriminative structure in the noise domain of NAF does not originate from the network itself?

- Would different types of noise affect the conclusions and experimental results?

---

> ### Author Response · Authors · 2025-11-25
>
> We sincerely appreciate your thoughtful review and valuable feedback. We address your comments point by point.
>
> >1. Are there any visualizations, experiments, or theoretical analyses demonstrating that the transferable discriminative structure in the noise domain of NAF does not originate from the network itself?
>
> **A1**: Thank you for your comment. As described in the implementation details in Appendix B2 of our submission, noise are projected into a representation space shared with the target representations only through a **single nonlinear layer** with ReLU activation. Accordingly, NAF does not feed noise into the ResNet backbone. The noise and target domains interact solely through alignment in the shared representation space. Therefore, the discriminative structure observed in the noise domain cannot be attributed to the backbone's inductive bias, **as the backbone never receives noise as input**. Instead, the discriminative structure arises from the predefined noise distribution in the noise space and from the supervised training applied to the noise representations in the representation space. We have included this clarification in the revision; please refer to Lines 307-310.
>
> >2. Would the noise distribution parameters (e.g., variance σ, dimensionality d) affect generalization performance?
>
> **A2**: We appreciate your suggestion. To evaluate the effectiveness of NAF under different noise generation strategies, we conduct additional experiments by varying the noise along three dimensions. (1) **Covariance Scale**: In the original setup, we first sample a mean for each class from a standard Gaussian distribution. Next, for each class, we generate individual noise from a Gaussian distribution with the corresponding mean and identity covariance $\mathbf{I}$. We additionally evaluate two configurations in which all class covariances are scaled to $0.1 \cdot \mathbf{I}$ and $10 \cdot \mathbf{I}$. (2) **Noise Dimensionality**: In the original setup, the noise dimensionality is set to 1024. We additionally evaluate two configurations with noise dimensionalities of 512 and 2048. (3) **Distribution Type**: In the original setup, the noise is drawn from a Gaussian distribution. We additionally test the Log-normal distribution and the Laplace distribution. The results, listed in the following table, indicate that NAF achieves comparable performance across a variety of noise settings, including variations in covariance scale, noise dimensionality, and distribution type. Those observations suggest that NAF can accommodate different noise configurations, highlighting its potential flexibility. We have included this experiment into Q10 in the revision.
>
> Table 1. Accuracy (%) comparison on CIFAR-100 using ResNet-18 with noise drawn from various noise generation strategies. Here, $\boldsymbol{\mu}_c$ is the class mean belonging to class $c$, and $d$ is the dimensionality of the noise.
> | Noise Configuration | Noise Distribution | Accuracy (%) |
> |---------------------------|-------------------------------------|--------------|
> | **Baseline**      | Gaussian: $\mathcal{N}(\boldsymbol{\mu}_c, \mathbf{I}), d = 1024$ | 49.98|
> | **Covariance Scale**    | Gaussian: $\mathcal{N}(\boldsymbol{\mu}_c, 0.1 \cdot \mathbf{I}), d = 1024$ | 50.38 |
> |                   | Gaussian: $\mathcal{N}(\boldsymbol{\mu}_c, 10 \cdot \mathbf{I}), d = 1024$ | 47.64  |
> | **Noise Dimensionality**| Gaussian: $\mathcal{N}(\boldsymbol{\mu}_c, \mathbf{I}), d = 512$ | 49.44   |
> |                   | Gaussian: $\mathcal{N}(\boldsymbol{\mu}_c, \mathbf{I}), d = 2048$ | 51.04  |
> | **Distribution Type**| Log-normal: $\log \mathcal{N}(\boldsymbol{\mu}_c, \mathbf{I}), d = 1024$ | 48.31 |
> |                      | Laplace: $\mathcal{L}((\boldsymbol{\mu}_c)_d, 1/\sqrt{2}), d = 1024$ | 49.99 |

---

> > ### Comment · Reviewer_Gj7X · 2025-11-26
> >
> > The rebuttal addresses my concerns. I will keep my rating and confidence.

---

> > > ### Author Response · Authors · 2025-11-26
> > >
> > > We sincerely thank you for your positive feedback and appreciate your time.

---

### Official Review · Reviewer_SonZ · 2025-10-27

**Soundness:** 1
**Presentation:** 2
**Contribution:** 2
**Rating:** 2
**Confidence:** 4

**Summary:**

This work propose a Noise Adaptation Framework (NAF), which introduces random Gaussian clusters as anchors to align unlabeled data in semi-supervised learning.

**Strengths:**

- the idea is simple and easy to follow

**Weaknesses:**

__Major Concerns:__
- theoretical justification
  - the proof of Theorem 1 is missing, and the derivation can be problematic.
  - rather than $d_{\mathcal{H}\Delta\mathcal{H}}(P_n,P_t)$, it should be $d_{\mathcal{H}\Delta\mathcal{H}}(D_n,D_t)$ that introduces the uncertainty.
  - the right-hand side should include a source error, i.e., $\epsilon_n(f)$
  - please provide the proof for: $\lambda\leq \hat{\lambda}$ since apparently some terms related to model complexity is missing.
  - $\epsilon_t(f)$ should not limited to labeled target data, which contradicts  line 243.
  - I strongly suggest the author check [1] proposing a theory for SSDA based on Ben-David (2010), which may help the derivation.
- algorithmic design
  - randomly sampled clusters from $\mathcal{N}(0,I)$ can overlap
  - the conditional discrepancy between randomly initialized Gaussian clusters and target features can be extremely large that cause substantial negative transfer.
  - the assumption that target feature must follow mixture of Gaussian distribution is too strick
- experiment results
  - the performance is far below the SOTA SSL methods such as DST
  - as for an increment to SSL, the comparison should be made between e.g., DST + LERM & DST + NAF


__Minor Concerns:__
- the code is not available from the provided link
- too much subscripts such as $u,l,n,e,t$, which is confusing

***
[1] Learning Invariant Representations and Risks for Semi-supervised Domain Adaptation, CVPR 2021

**Questions:**

see above

---

> ### Author Response · Authors · 2025-11-25
> **Part I**
>
> We sincerely appreciate your thoughtful review and valuable feedback. As suggested, we have revised the theorem’s statement to make it more explicit. For completeness, we first restate the key notations from Section 4.1 of the submission.
>
> Let $\mathcal{Z}$ be a domain-shared representation space, and let $\mathcal{F}$ be a hypothesis space over $\mathcal{Z}$, consisting of functions $f: \mathcal{Z} \rightarrow \\{0, 1\\}$ with VC dimension $d$. Denote by $\widetilde{\mathcal{P}}\_t$ and $\widetilde{\mathcal{P}}\_n$ the target and noise distributions over $\mathcal{Z}$, respectively. Let $\mathcal{U}\_t$, $\mathcal{U}\_n$ be unlabeled samples of size $m'$ each, drawn *i.i.d.* from $\widetilde{\mathcal{P}}\_t$ and $\widetilde{\mathcal{P}}\_n$, respectively. Let $\mathcal{L}\_t$ and $\mathcal{L}\_n$ be labeled samples of sizes $\beta m$ and $(1-\beta)m$, drawn *i.i.d.* from $\widetilde{\mathcal{P}}\_t$ and $\widetilde{\mathcal{P}}\_n$, respectively. Define $\hat{\epsilon}\_\alpha (f) = \alpha \hat{\epsilon}\_t (f) + (1 - \alpha) \hat{\epsilon}\_n (f)$ $(\alpha \in [0, 1])$ as the convex combination of the empirical target error $\hat{\epsilon}\_t (f)$ and empirical noise error $\hat{\epsilon}\_n (f)$, measured on $\mathcal{L}\_t$ and $\mathcal{L}\_n$, respectively. Based on those notations, we provide the revised Theorem 1.
>
> **Theorem 1**: Let $\hat{f} = \\arg\\min\_{f \in \mathcal{F}} \hat{\epsilon}\_{\alpha} (f)$ be the empirical minimizer of $\hat{\epsilon}\_\alpha (f)$, and let $f\_t\^\* = \\arg\\min\_{f \in \mathcal{F}} \epsilon\_t (f)$ be the target error minimizer. Then, for any $\delta \in (0, 1)$, with probability at least $1 - \delta$ (over the choice of the samples), we have:
> $$\begin{aligned}
> \epsilon\_t (\hat{f}) \leq & \epsilon\_t (f\_t\^\*) + 4 \sqrt{\frac{\alpha\^2}{\beta} + \frac{(1 - \alpha)\^2}{1 - \beta}} \sqrt{\frac{2d \log(2 (m + 1)) + 2 \log(\frac{8}{\delta})}{m}} \\\\ & + 2 (1 - \alpha) \Big( \frac{1}{2} \hat{d}\_{\mathcal{H}\Delta\mathcal{H}}(\mathcal{U}\_n, \mathcal{U}\_t) + 4\sqrt{\frac{2d \log(2m\') + \log\left(\frac{4}{\delta}\right)}{m\'}} + \hat{\epsilon}\_n (\hat{f}) + \hat{\epsilon}\_t (\hat{f}) + \varepsilon(d, \delta, (1 - \beta) m) + \varepsilon (d, \delta, \beta m) \Big),
> \end{aligned}$$
> where $\hat{d}\_{\mathcal{H} \Delta \mathcal{H}} (\mathcal{U}\_n, \mathcal{U}\_t)$ is the empirical $\mathcal{H}$-divergence estimated from noise and target samples in $\mathcal{Z}$, and $\varepsilon(d, \delta, \hat{m})$ denotes the complexity-dependent deviation term in the VC generalization bound (see, e.g., [1]), which depends on the VC dimension $d$ of the hypothesis space, the confidence parameter $\delta$, and the labeled sample size $\hat{m}$.
>
> [1] Vladimir N. Vapnik. Statistical Learning Theory. John Wiley & Sons, New York, 1998.
>
> Next, we address your comments point by point.

---

> ### Author Response · Authors · 2025-11-25
> **Part II**
>
> > 1. Theoretical justification: The proof of Theorem 1 is missing.
>
> **A1**: We appreciate your suggestion. Our analysis in Theorem 1 is primarily based on Theorem 3 of [2], and thus the proof was not included in the submission. We next provide its core proof steps. As a preliminary, we reference Lemmas 1 and 2, which correspond to Lemmas 4 and 5 in [2], respectively.
>
> **Lemma 1**: Let $f$ be a hypothesis in hypothesis space $\mathcal{F}$. Then $|\epsilon\_\alpha(f) - \epsilon\_t(f)| \le (1-\alpha)\left(\frac{1}{2} d\_{\mathcal{H}\Delta\mathcal{H}}(\widetilde{\mathcal{P}}\_n, \widetilde{\mathcal{P}}\_t) + \lambda \right),$
> where $\lambda := \min\_{f \in \mathcal{F}} \epsilon\_n (f) + \epsilon\_t (f)$.
>
> **Lemma 2**: For a fixed hypothesis $f$, if $m$ random labeled samples are drawn, with $\beta m$ from $\widetilde{\mathcal{P}}\_t$ and $(1 - \beta) m$ from $\widetilde{\mathcal{P}}\_n$, then for any $\delta \in (0, 1)$, with probability at least $1 - \delta$ (over the choice of samples), we have:
> $$
> \Pr\left[ \left| \hat{\epsilon}\_\alpha(f) - \epsilon\_\alpha(f) \right| \geq \epsilon \right] \leq 2 \exp\left( \frac{-2m\epsilon\^2}{\frac{\alpha\^2}{\beta} + \frac{(1-\alpha)\^2}{1 - \beta}} \right).
> $$
>
> For brevity, we omit the detailed proofs of Lemmas 1 and 2 here, which are available in [2]. Next, we provide a detailed proof for Theorem 1.
>
> *Proof*. In the proof below, steps labeled L1 and L2 correspond to applications of Lemma 1 and Lemma 2, respectively, with L2 additionally employing standard techniques of sample symmetrization and VC-dimension–based growth-function bounds [3].
> $$
> \begin{aligned}
> \epsilon\_t(\hat{f})
> &\le \epsilon\_\alpha(\hat{f}) + (1-\alpha) \left(\frac{1}{2} d\_{{\mathcal{H}\Delta \mathcal{H}}}(\widetilde{\mathcal{P}}\_n, \widetilde{\mathcal{P}}\_t) + \lambda \right) \quad \text{(L1)} \\\\
> &\le \hat{\epsilon}\_\alpha(\hat{f}) + 2 \sqrt{\frac{\alpha\^2}{\beta} + \frac{(1-\alpha)\^2}{1-\beta}} \sqrt{\frac{2d \log(2(m+1)) + 2 \log(\frac{8}{\delta})}{m}} \\\\
> &\quad + (1-\alpha) \left( \frac{1}{2} d\_{\mathcal{H}\Delta \mathcal{H}}(\widetilde{\mathcal{P}}\_n, \widetilde{\mathcal{P}}\_t) + \lambda \right) \quad \text{(L2)} \\\\
> &\le \hat{\epsilon}\_\alpha(f\_t\^\*) + 2 \sqrt{ \frac{\alpha\^2}{\beta} + \frac{(1-\alpha)\^2}{1-\beta} } \sqrt{\frac{2d \log(2(m+1)) + 2 \log(\frac{8}{\delta})}{m}} \\\\
> &\quad + (1-\alpha) \left(\frac{1}{2} d\_{\mathcal{H}\Delta \mathcal{H}}(\widetilde{\mathcal{P}}\_n, \widetilde{\mathcal{P}}\_t) + \lambda \right) \quad \left(\hat{f} = \\arg\\min\_{f \in \mathcal{F}} \hat{\epsilon}\_\alpha(f) \right) \\\\
> &\le \epsilon\_\alpha(f\_t\^\*) + 4 \sqrt{ \frac{\alpha\^2}{\beta} + \frac{(1-\alpha)\^2}{1-\beta} } \sqrt{\frac{2d \log(2(m+1)) + 2 \log(\frac{8}{\delta})}{m}} \\\\
> &\quad + (1-\alpha) \left( \frac{1}{2} d\_{\mathcal{H}\Delta \mathcal{H}}(\widetilde{\mathcal{P}}\_n, \widetilde{\mathcal{P}}\_t) + \lambda \right) \quad \text{(L2)} \\\\
> &\le \epsilon\_t(f\_t\^\*) + 4 \sqrt{\frac{\alpha\^2}{\beta} + \frac{(1-\alpha)\^2}{1-\beta} } \sqrt{\frac{2d \log(2(m+1)) + 2 \log(\frac{8}{\delta})}{m}} \\\\
> &\quad + 2(1-\alpha) \left(\frac{1}{2} d\_{\mathcal{H}\Delta \mathcal{H}}(\widetilde{\mathcal{P}}\_n, \widetilde{\mathcal{P}}\_t) + \lambda \right) \quad \text{(L1)} \\\\
> &\le \epsilon\_t(f\_t\^\*) + 4 \sqrt{\frac{\alpha\^2}{\beta} + \frac{(1-\alpha)\^2}{1-\beta}} \sqrt{\frac{2d \log(2(m+1)) + 2 \log(\frac{8}{\delta})}{m}} \\\\
> &\quad + 2(1-\alpha) \left( \frac{1}{2} d\_{\mathcal{H}\Delta \mathcal{H}}(\widetilde{\mathcal{P}}\_n, \widetilde{\mathcal{P}}\_t) + \epsilon\_n (\hat{f}) + \epsilon\_t (\hat{f}) \right) \quad \left( \lambda := \min\_{f \in \mathcal{F}} \epsilon\_n (f) + \epsilon\_t (f) \leq \epsilon\_n (\hat{f}) + \epsilon\_t (\hat{f}) \right) \\\\
> &\le \epsilon\_t(f\_t\^*\) + 4 \sqrt{\frac{\alpha\^2}{\beta} + \frac{(1-\alpha)\^2}{1-\beta}} \sqrt{\frac{2d \log(2(m+1)) + 2 \log(\frac{8}{\delta})}{m}} \\\\
> &\quad + 2(1-\alpha) \left(\frac{1}{2} d\_{\mathcal{H}\Delta \mathcal{H}}(\widetilde{\mathcal{P}}\_n, \widetilde{\mathcal{P}}\_t) + \hat{\epsilon}\_n (\hat{f}) + \hat{\epsilon}\_t (\hat{f}) + \varepsilon (d, \delta, (1 - \beta) m) + \varepsilon (d, \delta, \beta m) \right) \\\\
> &\le \epsilon\_t(f\_t\^\*) + 4 \sqrt{\frac{\alpha\^2}{\beta} + \frac{(1-\alpha)\^2}{1-\beta}} \sqrt{\frac{2d \log(2(m+1)) + 2 \log(\frac{8}{\delta})}{m}} \\\\
> &\quad + 2(1-\alpha) \left(\frac{1}{2} \hat{d}\_{\mathcal{H}\Delta\mathcal{H}}(\mathcal{U}\_n, \mathcal{U}\_t) + 4\sqrt{\frac{2d \log(2m\') + \log\left(\frac{4}{\delta}\right)}{m'}} + \hat{\epsilon}\_n (\hat{f}) + \hat{\epsilon}\_t (\hat{f}) + \varepsilon (d, \delta, (1 - \beta) m) + \varepsilon (d, \delta, \beta m) \right) \quad \left(\text{bound proposed in} [2] \right)
> \end{aligned}
> $$
> We have included this proof in Appendix C.2 in the revision.
>
> [2] Ben-David et al. A theory of learning from different domains. Machine learning, 2010.
>
> [3] Anthony, M., & Bartlett, P. (1999). Neural network learning: theoretical foundations.

---

> ### Author Response · Authors · 2025-11-25
> **Part III**
>
> > 2. Theoretical justification: Please provide the proof for: $\lambda \leq \hat{\lambda}$, since apparently some terms related to model complexity is missing.
>
> **A2**: We appreciate your comment. In the revised theorem, the dependence on $\hat{\lambda}$ has been eliminated, and all main conclusions remain valid.
>
> > 3. Theoretical justification: Rather than $d_{\mathcal{H}\Delta\mathcal{H}}(P_n, P_t)$, it should be $d_{\mathcal{H}\Delta\mathcal{H}}(D_n, D_t)$ that introduces the uncertainty.
>
> **A3**: Thank you for your suggestion. In the original theorem, we used the population divergence $d\_{\mathcal{H}\Delta\mathcal{H}}(\widetilde{\mathcal{P}}\_n, \widetilde{\mathcal{P}}\_t)$ for brevity. In the revised version, we adopt the empirical divergence $\hat{d}\_{\mathcal{H}\Delta\mathcal{H}}(\mathcal{U}\_n, \mathcal{U}\_t)$ as suggested.
>
> > 4. Theoretical justification: The right-hand side should include a source error, i.e., $\epsilon_n(f)$.
>
> **A4**: We appreciate your comment. In the revised theorem, we bound the expected noise error $\epsilon_n(\hat{f})$ by its empirical counterpart $\hat{\epsilon}_n(\hat{f})$ plus a complexity-dependent deviation term in the VC generalization bound $\varepsilon (d, \delta, (1 - \beta) m)$, yielding an empirical bound.
>
> > 5. Theoretical justification: $\epsilon_t(f)$ should not limited to labeled target data, which contradicts line 243.
>
> **A5**: Thank you for this comment. In the revised theorem, the expected target error $\epsilon_t(\hat{f})$ is bounded by its empirical counterpart $\hat{\epsilon}_t(\hat{f})$ plus a complexity-dependent deviation term in the VC generalization bound $\varepsilon (d, \delta, \beta m)$, yielding an empirical bound. Line 243 in the submission corresponds to $\hat{\epsilon}_t(\hat{f})$.
>
> > 6. Theoretical justification: I strongly suggest the author check [3] proposing a theory for SSDA based on Ben-David (2010), which may help the derivation.
>
> **A6**: Thank you for pointing us to [3]. Theorem 4.1 in [3] provides a practical bound using empirical errors from both domains while avoiding the joint optimal error term $\lambda$. In the revised theorem, we incorporate key insights from [3] into the framework of Theorem 3 in [1], yielding a bound that explicitly involves empirical noise and target errors as well as an empirical estimate of the distributional divergence without $\lambda$. We have cited [3] in the revision; please refer to Lines 236-237.
>
> [3] Li B et al. Learning invariant representations and risks for semi-supervised domain adaptation. CVPR, 2021.
>
> > 7. Algorithmic design: Randomly sampled clusters from $\mathcal{N} (0, \mathbf{I})$ can overlap
>
> **A7**: Thank you for your fededback. In our algorithm design, we take two measures to maintain separability of the noise in the representation space: (1) we first sample $C$ class means from $\mathcal{N} (0, \mathbf{I})$ in a high-dimensional space (e.g., 1024), and then generate noise around each class mean using a Gaussian distribution with identity covariance, which potentially reduces the possibility of overlap, and (2) all noise is labeled and optimized via empirical risk minimization, encouraging distinguishable clusters in the representation space. Those together help ensure that noise from different classes remains separable.
>
> > 8. Algorithmic design: The conditional discrepancy between randomly initialized Gaussian clusters and target features can be extremely large that cause substantial negative transfer.
>
> **A8**: Thank you for your suggestion. While the Gaussian clusters are randomly initialized in the input space, **NAF operates in a domain-shared representation space**, where the model jointly minimizes the empirical errors of both domains and reduces their distributional divergence. In NAF, a concrete instantiation of the distribution alignment module is NDS, which aims to align the global and class-wise means of both domains in the representation space. This helps mitigate the conditional discrepancy between domains. Empirically, we observe consistent positive transfer across most scenarios, indicating the effectiveness of the proposed NAF.

---

> ### Author Response · Authors · 2025-11-25
> **Part IV**
>
> > 9. Algorithmic design: the assumption that target feature must follow mixture of Gaussian distribution is too strict
>
> **A9**: we appreciate your comment. We clarify that NAF does not assume that the target features follow a mixture of Gaussian distributions. In our experiments, Gaussian mixture distributions are simply used as one instantiation of the noise generation mechanism. The NAF framework itself is compatible with other noise types, such as Laplace and log-normal distributions. As shown in the table below, NAF achieves comparable performance across different noise types. Those results suggest that NAF potentially accommodates distinct noise types. We have included this experiment into Q10 in the revision.
>
> Table 1. Accuracy (%) comparison on CIFAR-100 using ResNet-18 with noise drawn from distinct noise types. Here, $\boldsymbol{\mu}\_c$ is the class mean belonging to class $c$, and $d$ is the dimensionality of the noise.
> |Noise Type|Accuracy|
> |-|-|
> | Gaussian: $\mathcal{N}(\boldsymbol{\mu}_c, \mathbf{I}), d = 1024$ | 49.98|
> | Log-normal: $\log \mathcal{N}(\boldsymbol{\mu}_c, \mathbf{I}), d = 1024$ | 48.31 |
> | Laplace: $\mathcal{L}((\boldsymbol{\mu}_c)_d, 1/\sqrt{2}), d = 1024$ | 49.99 |
>
> > 10. Experiment results: The performance is far below the SOTA SSL methods such as DST
>
> **A10**: Thank you for your feedback. Our method is not designed as a standalone SSL competitor to well-established methods such as DST. Instead, its core contribution lies in introducing an external synthetic noise domain for adaptation. Our experiments demonstrate that leveraging such a noise domain can improve adaptation performance, and that **our method can provide additional gains to exsiting SSL methods like DST**. Overall, our work focuses on exploring how incorporating a noise domain can facilitate knowledge transfer.
>
> > 11. Experiment results: As for an increment to SSL, the comparison should be made between e.g., DST + LERM & DST + NAF
>
> **A11**: We appreciate your suggestion. We have conducted additional experiments on CIFAR-10 using ResNet-18 comparing NAF and LERM under both ERM and DST. As shown in the table below, NAF brings a larger improvement than LERM when built upon ERM. On DST, both methods provide modest improvements, and NAF shows comparable performance to LERM. Those results suggest that NAF may serve as a competitive plug-in module for SSL. We have included this experiment into Q18 in the revision.
>
> Table 2. Accuracy (%) comparison of ERM and DST combined with either LERM or NAF on CIFAR-10 using ResNet-18.
> |Method|Base|+ LERM|+ NAF|
> |-|-|-|-|
> | ERM | 58.15 | 64.90 | 71.83 |
> | DST | 84.96 | 86.82 | 86.58 |
>
> > 12. Minor Concerns: The code is not available from the provided link. Too much subscripts such as $u, l, n, e, t$, which is confusing
>
> **A12**: Thank you for your feedback. We have confirmed that our anonymous repository is accessible. Additionally, since this work involves different types of data, we use the notations $u, l, t, e, n$ to distinguish them: $u$ for unlabeled target samples, $l$ for labeled target samples, $e$ for test target samples, $t$ for all target samples, and $n$ for noise. For clarity, we have included the following notation table in Appendix C.1 in the revision.
>
> Table 3. A summary of the notations used in this paper.
> |Notation|Description|
> |-|-|
> | $C$ | Total number of classes |
> | $\mathcal{C}$ | Class index set {$0, \dots, C-1$} |
> | $\mathcal{D}\_l$, $\mathcal{D}\_u$ | Labeled and unlabeled target sample set |
> | $\mathcal{D}\_e$ |  Test target sample set (used only for evaluation) |
> | $\mathcal{D}\_t$ | Target domain: $\mathcal{D}\_l \cup \mathcal{D}\_u \cup \mathcal{D}\_e$ |
> | $\mathcal{D}\_n$ | Noise domain |
> | $\mathbf{x}\_i^l, \mathbf{x}\_i\^u, \mathbf{x}\_i\^e$ | $i$-th sample from $\mathcal{D}\_l$, $\mathcal{D}\_u$, and $\mathcal{D}_e$, respectively |
> | $y\_i\^l$ | Label of $\mathbf{x}\_i\^l$, $y\_i\^l \in \mathcal{C}$ |
> | $\mathbf{n}\_i$ | $i$-th noise in $\mathcal{D}\_n$ |
> | $y\_i$ | Label of $\mathbf{n}\_i$, $y\_i \in \mathcal{C}$ |
> | $\mathcal{X}$ | Sample space (e.g., a pixel-level image space) |
> | $\mathcal{E}$ | Noise space (e.g., a $p$-dimensional space) |
> | $\mathcal{Z}$ | Domain-shared representation space |
> | $\mathcal{F}$ | Hypthesis space over $\mathcal{Z}$ |
> | $\widetilde{\mathcal{P}}\_t$ | Target distribution over $\mathcal{Z}$ |
> | $\widetilde{\mathcal{P}}\_n$  | Noise distribution over $\mathcal{Z}$ |
> | $\mathcal{U}\_n$, $\mathcal{U}\_t$ | Unlabeled sample sets drawn from $\widetilde{\mathcal{P}}\_n$ and $\widetilde{\mathcal{P}}\_t$, respectively |
> | $\mathcal{L}\_n$, $\mathcal{L}\_t$ | Labeled sample sets drawn from $\widetilde{\mathcal{P}}\_n$ and $\widetilde{\mathcal{P}}_t$, respectively |
> | $g\_t (\cdot)$ | Representation extractor for target samples |
> | $g\_n (\cdot)$ | Noise projector for noise |
> | $f (\cdot)$ | Domain-shared classifier |
> | $n\_l$, $n\_u$ | Number of labeled and unlabeled target samples |
> | $n$ | Number of noise |

---

> > ### Comment · Reviewer_SonZ · 2025-11-28
> > **response**
> >
> > I appreciate the detailed explanation from the author. However, I am not convinced. My concerns still largely remains in the following two aspects:
> > ****
> >
> > __Theoretical Justification__
> >
> >  - The theorem has been substantially revised regarding some incorrect statements such as $\lambda \leq \hat{\lambda}$.
> >  - However, the provided theorem has __limited novelty__ compared to Theorem 3 in Ben-David's (2010), which simply substitutes the source $S$ with the noise domain.
> >  - In addition, the theorem is clearly __not informative__ enough to justify the proposed algorithm, as the mechanism of how the introduced noise could reduce the generalization error on target domain remains unexplained.
> >  - Let me elaborate a bit more. In the last step, the author further upper bound $\lambda$ by:
> >
> > $\lambda\leq \epsilon_t(\hat{f})+\epsilon_n(\hat{f}) \leq \hat{\epsilon}_t(\hat{f}) +\mathcal{O}\sqrt{\frac{d}{\beta m}\log \frac{\beta m}{d}+\frac{1}{\beta m}\log \frac{1}{\delta}}+\epsilon_n(\hat{f})$, where most of the generalization error still comes from the limited size of the labeled target data in SSL since the author mentions the empirical target error $\hat{\epsilon}_t(\hat{f})$ is measured only on labeled data $\mathcal{L}_t$. Therefore, the entire bound becomes non-informative and the divergence related to the noise can be considered redundant since we can always derive the following from the beginning:
> > $\epsilon_t(\hat{f}) \leq \hat{\epsilon}_t(\hat{f}) +\mathcal{O}\sqrt{\frac{d}{\beta m}\log \frac{\beta m}{d}+\frac{1}{\beta m}\log \frac{1}{\delta}}$
> >  - As for a minor concern, the revised proof seems still not rigorous. E.g., in the last line, it should be $\log \frac{8}{\delta}$ according to Ben-David's (2010); the coefficient related to $\delta$ should be modified since the author introduce an additional inequality between $\lambda$ and $\hat{\epsilon}_t(\hat{f}) +\hat{\epsilon}_n(\hat{f}) $, which requires the union bound to make sure all the inequalities hold.
> >
> > __Empirical Importance__
> >  - The proposed algorithm is pure incremental rather than a powerful SSL techniques such as DST, which makes it difficult to assess its empirical value.
> >  - Since in practice, people will not choose ERM over DST as their primary SSL regularization due to the large performance gap, which makes the gain by NAF less meaningful. Moreover, the marginal difference compared to LERM for DST base, which I assume is the current SOTA of SSL, further lowers down the necessity of deploying NAF when dealing with real-world problems.

---

> ### Author Response · Authors · 2025-12-03
> **Part I**
>
> We sincerely thank you for your detailed comments and valuable suggestions. Below, we address your comments point by point.
>
> >1. Theoretical Justification: the provided theorem has limited novelty compared to Theorem 3 in Ben-David's (2010), which simply substitutes the source with the noise domain. In addition, the theorem is clearly not informative enough to justify the proposed algorithm, as the mechanism of how the introduced noise could reduce the generalization error on target domain remains unexplained.
>
> **A1**. Thank you for your comment. While our theory builds upon semi-supervised transfer learning (SSTL), it offers two distinct perspectives. First, our theoretical analysis suggests that, **regardless of the origin of the source domain** (e.g., images, text, or synthetic noise), the generalization bound on the expected target error can be tightened when empirical errors on both domains and their empirical distributional divergence are effectively minimized in a shared representation space. Second, it **relaxes the common assumption** in SSTL that source and target domains are related, which **explains why even a synthetic noise domain can serve as an effective surrogate in practice**. Those insights provide theoretical support for the design of the proposed NAF framework. We have included this clarification into Section 4.1 of the revision.
>
> >2. Theoretical Justification: Let me elaborate a bit more. In the last step, the author further upper-bounds $\lambda$ by: $\lambda \le \epsilon\_t(\hat{f}) + \epsilon\_n(\hat{f}) \le \hat{\epsilon}\_t(\hat{f}) + \mathcal{O} \left( \sqrt{\frac{d}{\beta m} \log \frac{\beta m}{d} + \frac{1}{\beta m} \log \frac{1}{\delta}} \right) + \epsilon\_n(\hat{f}),$ where most of the generalization error still comes from the limited size of the labeled target data in SSL, since the author mentions that the empirical target error $\hat{\epsilon}\_t(\hat{f})$ is measured only on labeled data $\mathcal{L}\_t$. Therefore, the entire bound becomes non-informative, and the divergence related to the noise can be considered redundant since we can always derive the following from the beginning:
> $\epsilon\_t(\hat{f}) \le \hat{\epsilon}\_t(\hat{f}) + \mathcal{O} \left(\sqrt{\frac{d}{\beta m} \log \frac{\beta m}{d} + \frac{1}{\beta m} \log \frac{1}{\delta}} \right).$
>
> **A2**: We appreciate your thoughtful analysis. The reviewer’s derivation focuses solely on the target-domain component, treating the problem as a **single-domain** case. In contrast, Theorem 1 is derived in a **two-domain** setting, where the generalization bound depends on three key terms: (1) the empirical target error, (2) the empirical noise error, and (3) their empirical distributional divergence. Tightening the bound thus requires controlling all three terms, which is exactly what NAF optimizes. If either the empirical noise error or the divergence remains large, the bound on $\epsilon_t (\hat{f})$ cannot be effectively tightened, even when $\hat{\epsilon}_t (\hat{f})$ is small. This is consistent with our empirical results: as shown in Figure 4, ERM (which only minimizes $\hat{\epsilon}_t (\hat{f})$) performs worse than NAF. Therefore, the noise domain is not redundant. We have included this clarification into Section 4.1 in the revision.

---

> ### Author Response · Authors · 2025-12-03
> **Part II**
>
> > 3. As for a minor concern, the revised proof seems still not rigorous. E.g., in the last line, it should be $\log \frac{8}{\delta}$ according to Ben-David's (2010); the coefficient related to should be modified since the author introduce an additional inequality between $\lambda$ and $\hat{\epsilon}\_t (\hat{f}) + \hat{\epsilon}\_n (\hat{f})$, which requires the union bound to make sure all the inequalities hold.
>
> **A3**: We appreciate your suggestion. In [1], the second complexity term in Theorem 3 is written as $\log \tfrac{8}{\delta}$ in the main text, while Section 6 and the appendix consistently use $\log \tfrac{4}{\delta}$. We think that the latter one is correct, as it derives from applying a union bound over two events starting from $\log \frac{2}{\delta}$ in Theorem 2 of [1].
>
> Our analysis introduces two additional high-probability events that control the deviations between expected and empirical errors: one for the target domain and another for the noise domain. To ensure that the overall failure probability does not exceed $\delta$, we apply a union bound and allocate $\frac{\delta}{4}$ to each event. This adjustment leads to the logarithmic factors $\log \frac{16}{\delta}$ and $\log \frac{8}{\delta}$ in the first and second complexity terms of Theorem 1, respectively. For clarity, we present the asymptotic form of the bound in the main text and provide the full non-asymptotic expression in the proof in Appendix C.2 of the revision. Accordingly, Theorem 1 has been updated to:
> $$
> \begin{aligned}
> \epsilon\_t (\hat{f}) &\leq \epsilon\_t (f\_t\^*) + \mathcal{O} \left( \sqrt{ \frac{\alpha\^2}{\beta} + \frac{(1 - \alpha)\^2}{1 - \beta} } \sqrt{ \frac{d \log m + \log(\frac{1}{\delta})}{m} } \right) + 2(1 - \alpha) \Bigg[ \frac{1}{2} \hat{d}\_{\mathcal{H}\Delta\mathcal{H}}(\mathcal{U}\_n, \mathcal{U}\_t) + \mathcal{O} \left( \sqrt{\frac{d \log m\' + \log(\frac{1}{\delta})}{m\'} } \right) \\\\ &\quad + \hat{\epsilon}\_n (\hat{f}) + \hat{\epsilon}\_t (\hat{f}) + \mathcal{O} \left( \sqrt{ \frac{d \log(\frac{(1-\beta)m}{d}) + \log(\frac{1}{\delta})}{(1-\beta)m} } \right) + \mathcal{O} \left( \sqrt{ \frac{d \log(\frac{\beta m}{d}) + \log(\frac{1}{\delta})}{\beta m} } \right) \Bigg].
> \end{aligned}
> $$
>
> [1] Ben-David et al. A theory of learning from different domains[J]. Machine learning, 2010, 79(1): 151-175.
>
> >4. Empirical Importance: The proposed algorithm is pure incremental rather than a powerful SSL techniques such as DST, which makes it difficult to assess its empirical value. Since in practice, people will not choose ERM over DST as their primary SSL regularization due to the large performance gap, which makes the gain by NAF less meaningful. Moreover, the marginal difference compared to LERM for DST base, which I assume is the current SOTA of SSL, further lowers down the necessity of deploying NAF when dealing with real-world problems.
>
> **A4**: Thank you for your comment. We would like to clarify that our focus is on investigating the role of domain-agnostic noise, rather than on proposing a SOTA SSL method (e.g., DST) or plug-in (e.g., LERM). Our experiments provide an initial exploration of this idea on standard benchmarks, showing that NAF can effectively improve both ERM and SOTA SSL baselines. Those results suggest that domain-agnostic noise could serve as a useful surrogate and a starting point for studying its role in transfer learning.

---

### Official Review · Reviewer_L2yW · 2025-11-01

**Soundness:** 3
**Presentation:** 2
**Contribution:** 3
**Rating:** 6
**Confidence:** 4

**Summary:**

This paper introduced a transfer learning based theoretical framework that learn from the noise domain to help the target domain in a semi-supervised learning manner.

**Strengths:**

1. The author formulated a novel theoretical framework to upper-bound the semi-supervised domain adaptation, interestingly, the source domain is a noise domain, and this paper is well-motivated.
2. The method origninated from the theretical framework is reasonable.
3. The experiment is good.

**Weaknesses:**

1. The underlying assumption of this paper is Gaussian distribution, so the author may discuss some other noise distribution settings.
2. I think the success of the knowledge transfer from the source to target is, the class is actually seperable from the source domain, e.g., the author is actually transfering the seperable characteristic of the noise source domain to the target domain (from the Figs of the main paper). Therefore, one concern arises that, for the Gaussian which generates noise in the source domain, will the different sampling strategy would the performance drops significantly? And whether the distance of the sampled Gaussians would affect the performance.
3. Did you experiemnt on the domain adaptation dataset? e.g., VisDA? Since your contribution is the knowledge transfer, why you only employ the CV tasks.
4. From Fig.4, it seems the $\mathcal{L}_n$ is the main contribution to lead to the performance improvement, but how you construct the label of the source noise domains? Could you be more clear on this point?

**Questions:**

See weakness

---

> ### Author Response · Authors · 2025-11-25
> **Part I**
>
> We sincerely appreciate your thoughtful review and valuable feedback. We address your comments point by point.
>
> > 1. The underlying assumption of this paper is Gaussian distribution, so the author may discuss some other noise distribution settings. For the Gaussian which generates noise in the source domain, will the different sampling strategy would the performance drops significantly?
>
> **A1**: Thank you for your suggestion. To evaluate the effectiveness of NAF under different noise generation strategies, we conduct additional experiments by varying the noise along three dimensions. (1) **Covariance Scale**: In the original setup, we first sample a mean for each class from a standard Gaussian distribution. Next, for each class, we generate individual noise from a Gaussian distribution with the corresponding mean and identity covariance $\mathbf{I}$. We additionally evaluate two configurations in which all class covariances are scaled to $0.1 \cdot \mathbf{I}$ and $10 \cdot \mathbf{I}$. (2) **Noise Dimensionality**: In the original setup, the noise dimensionality is set to 1024. We additionally evaluate two configurations with noise dimensionalities of 512 and 2048. (3) **Distribution Type**: In the original setup, the noise is drawn from a Gaussian distribution. We additionally test the Log-normal distribution and the Laplace distribution. The results, listed in the following table, indicate that NAF achieves comparable performance across a variety of noise settings, including variations in covariance scale, noise dimensionality, and distribution type. Those observations suggest that NAF can accommodate different noise configurations, highlighting its potential flexibility. We have included this experiment into Q10 in the revision.
>
> Table 1. Accuracy (%) comparison on CIFAR-100 using ResNet-18 with noise drawn from various noise generation strategies. Here, $\boldsymbol{\mu}_c$ is the class mean belonging to class $c$, and $d$ is the dimensionality of the noise.
> | Noise Configuration | Noise Distribution | Accuracy (%) |
> |-|-|-|
> | **Baseline**      | Gaussian: $\mathcal{N}(\boldsymbol{\mu}_c, \mathbf{I}), d = 1024$ | 49.98|
> | **Covariance Scale**    | Gaussian: $\mathcal{N}(\boldsymbol{\mu}_c, 0.1 \cdot \mathbf{I}), d = 1024$ | 50.38 |
> |                   | Gaussian: $\mathcal{N}(\boldsymbol{\mu}_c, 10 \cdot \mathbf{I}), d = 1024$ | 47.64  |
> | **Noise Dimensionality**| Gaussian: $\mathcal{N}(\boldsymbol{\mu}_c, \mathbf{I}), d = 512$ | 49.44   |
> |                   | Gaussian: $\mathcal{N}(\boldsymbol{\mu}_c, \mathbf{I}), d = 2048$ | 51.04  |
> | **Distribution Type**| Log-normal: $\log \mathcal{N}(\boldsymbol{\mu}_c, \mathbf{I}), d = 1024$ | 48.31 |
> |                      | Laplace: $\mathcal{L}((\boldsymbol{\mu}_c)_d, 1/\sqrt{2}), d = 1024$ | 49.99 |
>
> > 2. Whether the distance of the sampled Gaussians would affect the performance.
>
> **A2**: We appreciate your suggestion. We perform additional ablation studies by constructing noise domains with controlled inter-class distances. Specifically, we first sample a global mean $\boldsymbol{\mu}$ and class-specific offsets $\boldsymbol{\epsilon}_c$ from a standard Gaussian distribution, and define class means as $\boldsymbol{\mu}_c = \boldsymbol{\mu} + \delta \boldsymbol{\epsilon}_c$, where $\delta$ explicitly controls the distance between class means. Then, we sample 50 noise per class from the Gaussian distribution $\mathcal{N}(\boldsymbol{\mu}_c, \mathbf{I})$. By varying $\delta$ over the set {$0, 0.1, 0.3, 0.5, 1$}, we adjust the inter-class distances of the noise domain, corresponding to Jensen–Shannon (JS) divergence values of 0, 2.57, 23.16, 64.33, and 257.33, respectively. Higher JS divergence values indicate larger inter-class distances. The results on CIFAR-100 using ResNet-18 are reported in Table 2. We can see that when $\delta = 0$, NAF performs comparably to Empirical Risk Minimization (ERM). As the value of $\delta$ increases, the performance improves accordingly, indicating that larger inter-class distances in the noise domain lead to enhanced generalization performance. We have included this experiment into Q17 in the appendix of the revision.
>
> Table 2. Accuracy (%) comparison on CIFAR-100 using ResNet-18 with varying inter-class distances of the noise domain.
>
> | $\delta$ | 0     | 0.1   | 0.3   | 0.5   | 1     |
> |---------|-------|-------|-------|-------|-------|
> | **ERM** | 42.24 | 42.24 | 42.24 | 42.24 | 42.24 |
> | **NAF** | 43.80 | 43.78 | 46.57 | 49.57 | 49.78 |

---

> ### Author Response · Authors · 2025-11-25
> **Part II**
>
> > 3. Did you experiemnt on the domain adaptation dataset? e.g., VisDA?
>
> **A3**: Yes, we have conducted experiments on domain adaptation datasets. As detailed in Q9 of the submission, we evaluated NAF on the Office-Caltech-10 dataset, using Caltech-10 as the target domain with 4 labeled samples per class for training and the remaining samples treated as unlabeled. We compare NAF (Noise), which uses the synthetic noise domain as the source domain, with NAF (Real), which uses the Amazon domain as the source domain. For each source domain, we also vary the number of labeled samples per class among 10, 20, 30, 40, and 50. As shown in Table 3, both NAF (Noise) and NAF (Real) outperform ERM, with NAF (Real) performing slightly better. Those results indicate that synthetic noise can serve as a practical substitute when real out-of-domain samples are unavailable. Detailed descriptions are provided in Q9 of submission (corresponding to Q13 in the revision).
>
> Table 3. Accuracy (%) comparison on Amazon-to-Caltech-10 transfer task using ResNet-18 with different number of source samples.
> | # source samples per class | 10    | 20    | 30    | 40    | 50    |
> |----------------------------|-------|-------|-------|-------|-------|
> | ERM                        | 83.51 | 83.51 | 83.51 | 83.51 | 83.51 |
> | NAF (Noise)                | 89.89 | 88.65 | 88.83 | 88.12 | 89.36 |
> | NAF (Real)                 | 90.25 | 90.07 | 90.96 | 92.20 | 91.14 |
>
>
> > 4. Since your contribution is the knowledge transfer, why you only employ the CV tasks.
>
> **A4**: Thank you for your comment. To evaluate NAF on non-visual tasks, we conduct experiments on the AG News dataset, which contains news articles from four topics. We randomly sample four labeled and 1000 unlabeled target samples, and generate 50 noise per class. Target texts are encoded using a pre-trained BERT model, and noise is mapped through a nonlinear projector with ReLU activation. NAF achieves an accuracy of 82.82%, outperforming ERM, which achieves 78.64%. The results suggest that NAF could potentially facilitate knowledge transfer in non-visual tasks. We have included this experiment into Q4 in the revision.
>
> > 5. From Fig.4, it seems $\mathcal{L}_n$ is the main contribution to lead to the performance improvement, but how you construct the label of the source noise domains? Could you be more clear on this point?
>
> **A5:** Thank you for your suggestion. In our implementation, each target class is first assigned a distinct numeric index in {0, $\cdots$, $C-1$} with $C$ classes (e.g., $C = 10$ for CIFAR-10). We then set the number of noise classes equal to the number of target classes $C$ and, for each class, sample 50 noise from a distinct Gaussian distribution. **All noise drawn from each Gaussian distribution is assigned a distinct numeric index in {$0, \dots, C-1$} prior to training**. This ensures that noise from different Gaussian distributions is mapped to different class indices, establishing a fixed one-to-one correspondence between noise and target classes. We have included this clarification in the revision; please refer to Lines 172–176.

---

### Author Response · Authors · 2025-12-03
**General Response**

We sincerely appreciate the reviewers' insightful comments. Below, we present a brief summary of our rebuttal.

First, we thank the reviewers for their encouraging comments.

**Reviewer L2yW** notes the novel theoretical framework, clear motivation, sound method design, and good experimental results.

**Reviewer SonZ** appreciates that the idea is simple and easy to follow.

**Reviewer Gj7X** comments the SSNA problem for its potential to use the noise domain as a surrogate in privacy or data-scarcity scenarios, and for providing an explicit generalization bound and reproducible code.

**Reviewer GdmS** praises the novel idea of transferring structure from a synthetic noise domain, the connection to domain adaptation bounds, extensive experiments, as well as clear writing and reproducible code.

Then, we address the following concerns.

To **Reviewer L2yW**: We have investigated various noise generation strategies (`Q10`) and controlled inter-class distances (`Q17`), evaluated on the Office-Caltech-10 (`Q13`) and AG News datasets (`Q4`), and clarified the label construction of the noise (`Section 4.1`).

To **Reviewer SonZ**: We have revised Theorem 1, clarified the theoretical contributions (`Section 4.1`), explained the algorithm design, and included a notation table (`Appendix C.1`). We also clarified that our core contribution is to investigate whether a domain-agnostic noise domain can serve as a useful surrogate for the target task, rather than proposing a SOTA semi-supervised learning method or plug-in.

To **Reviewer Gj7X**: We have clarified the origin of the discriminative structure in the noise domain (`Section 4.2`) and investigated various noise generation strategies (`Q10`).

To **Reviewer GdmS**: We have redesigned Figure 1, investigated class imbalance in the target domain (`Q11`), explored various noise generation strategies (`Q10`) and a contrastive learning method (`Q19`), and clarified the effect of the noise domain (`Section 4.1 `&`Q9`).

In summary, those clarifications have improved the manuscript, and we hope they adequately address the reviewers’ concerns. Thanks again.

---

### Meta-Review · Area_Chair_XSui · 2025-12-31

**Summary:**

The paper provides a theoretical analysis of what the authors coin, semi-supervised noise adaptation (SSNA), a semi-supervised transfer learning setting in which a noise-to-target domain transfer is performed. The central contribution of the work is Theorem 1, which provides a generalisation bound for SSNA (under a Gaussianity assumption). The authors then provide several experiments and ablations.

**Reviewer Concerns:**

Reviewer SonZ has expressed concerns about the theoretical validity, as the initial version lacked a proof of Theorem 1. The authors then revised the theorem during the rebuttal phase and provided a proof of the revised theorem. As this work is of a theoretical nature and hence requires a rigorous theoretical justification of the results, it is prudent to reassess the new theoretical result (the central contribution) presented during the rebuttal. This necessitates a resubmission of the work to a future venue.

**Reviewer Scores:**

The review scores have indicated a large spread for this submission. Here is a potential change in the scores.

- L2yW: I expect the reviewer to keep the score, but it is possible the score might have dropped as a result of the discussion with SonZ.

- SonZ: The reviewer has expressed remaining concerns after the rebuttal of the authors. I believe the assessment is accurate and warrants rejecting the submission.

- Gj7X: I expect the reviewer to keep the score, but it is possible the score might have dropped as a result of the discussion with SonZ.

- GdmS: The reviewer expressed having kept the score.

---

### Decision · Program_Chairs · 2026-01-26

Reject